# Mitochondrial apolipoprotein MIC26 is a metabolic rheostat regulating central cellular fuel pathways

Melissa Damiecki[1], Ritam Naha[1], Yulia Schaumkessel[1], Philipp Westhoff[2,3], Nika Atanelov[1], Anja Stefanski[4], Patrick Petzsch[5], Kai Stühler[4,6], Karl Köhrer[5], Andreas PM Weber[2,3], Ruchika Anand[1], Andreas S Reichert[1], Arun Kumar Kondadi[1]

**Mitochondria play central roles in metabolism and metabolic disorders such as type 2 diabetes. MIC26, a mitochondrial contact site and cristae organising system complex subunit, was linked to diabetes and modulation of lipid metabolism. Yet, the functional role of MIC26 in regulating metabolism under hyperglycemia is not understood. We used a multi-omics approach combined with functional assays using WT and *MIC26* KO cells cultured in normoglycemia or hyperglycemia, mimicking altered nutrient availability. We show that MIC26 has an inhibitory role in glycolysis and cholesterol/lipid metabolism under normoglycemic conditions. Under hyperglycemia, this inhibitory role is reversed demonstrating that MIC26 is critical for metabolic adaptations. This is partially mediated by alterations of mitochondrial metabolite transporters. Furthermore, *MIC26* deletion led to a major metabolic rewiring of glutamine use and oxidative phosphorylation. We propose that MIC26 acts as a metabolic "rheostat," that modulates mitochondrial metabolite exchange via regulating mitochondrial cristae, allowing cells to cope with nutrient overload.**

## Introduction

The increasing prevalence of global obesity is a huge biological risk factor for development of a range of chronic diseases including cardiovascular diseases, musculoskeletal and metabolic disorders (GBD 2015 Obesity Collaborators et al, 2017). At the cellular level, obesity is associated with DNA damage, inflammation, oxidative stress, lipid accumulation, and mitochondrial dysfunction (Włodarczyk & Nowicka, 2019). Mitochondria play

central roles in anabolic and catabolic pathways (Spinelli & Haigis, 2018) and as a consequence mitochondrial dysfunction is associated with a variety of metabolic diseases such as type 2 diabetes mellitus (T2DM) (Szendroedi et al, 2011). Mitochondrial dysfunction is often linked to abnormal mitochondrial ultrastructure (Zick et al, 2009; Kondadi et al, 2020b; Eramo et al, 2020; Kondadi & Reichert, 2024) and abnormal mitochondrial ultrastructure was also associated with diabetes (Bugger et al, 2008; Xiang et al, 2020). Mitochondria harbour two membranes, the mitochondrial outer membrane (OM) and the inner membrane (IM). The part of the IM closely apposed to the OM is termed the inner boundary membrane whereas the IM which invaginates towards the mitochondrial matrix is termed the cristae membrane (CM). Crista junctions (CJs) are pore-like structures around 12–25 nm in diameter, separating the inner boundary membrane and CM and are proposed to act as diffusion barriers for proteins and metabolites (Frey & Mannella, 2000; Mannella et al, 2013). Formation of CJs depends on Mic60 (Fcj1, Mitofilin, IMMT) which was shown to be located at CJs regulating cristae formation in concert with the $F_1F_O$ ATP synthase (Zick et al, 2009). Mic60 is a subunit of the "mitochondrial contact site and cristae organising system" (MICOS) complex (Harner et al, 2011; Hoppins et al, 2011; von der Malsburg et al, 2011) which consists of seven proteins organised into two subcomplexes: MIC60/MIC19/MIC25 and MIC10/MIC26/MIC27 with MIC13 stabilizing the MIC10-subcomplex in mammals (Guarani et al, 2015; Anand et al, 2016; Urbach et al, 2021). MIC26/APOO harbours an apolipoprotein A1/A4/E family domain and therefore was classified as an apolipoprotein (Lamant et al, 2006; Koob et al, 2015). Traditionally, apolipoproteins mediate lipid and cholesterol metabolism by facilitating the formation of lipoproteins and regulating their distribution to different tissues via the blood stream (Mehta & Shapiro, 2022). Initially, MIC26 was identified as a protein of unknown function in cardiac transcriptome of dogs fed with high-fat diet (HFD) (Philip-

[1]Institute of Biochemistry and Molecular Biology I, Medical Faculty and University Hospital Düsseldorf, Heinrich Heine University Düsseldorf, Düsseldorf, Germany [2]Institute of Plant Biochemistry, Cluster of Excellence on Plant Sciences (CEPLAS), Heinrich Heine University Düsseldorf, Düsseldorf, Germany [3]Plant Metabolism and Metabolomics Laboratory, Cluster of Excellence on Plant Sciences (CEPLAS), Heinrich Heine University, Düsseldorf, Germany [4]Molecular Proteomics Laboratory, Medical Faculty and University Hospital, Heinrich Heine University Düsseldorf, Düsseldorf, Germany [5]Genomics and Transcriptomics Laboratory, BMFZ, Heinrich Heine University Düsseldorf, Düsseldorf, Germany [6]Institute of Molecular Medicine, Protein Research, Medical Faculty and University Hospital, Heinrich Heine University Düsseldorf, Düsseldorf, Germany

Correspondence: kondadi@hhu.de

Couderc et al, 2003) and was incorrectly assumed to exist as a 55 kD O-linked glycosylated protein as it was immunopositive to a custom-generated MIC26 antibody in samples of human serum, heart tissue and HepG2 cell line (Lamant et al, 2006). However, the recombinant protein was only observed at the expected size of 22 kD (Lamant et al, 2006) and it was later shown that this 22 kD form is located to mitochondria (Koob et al, 2015; Ott et al, 2015). Moreover, using several cellular MIC26 deletion models and different antibodies, we showed recently that MIC26 is exclusively present as a 22 kD mitochondrial protein and not as a 55 kD protein (Lubeck et al, 2023). In light of these findings, the primary physiological function of MIC26 in diabetes is linked to its role in the mitochondrial IM and not to an earlier proposed secreted form of MIC26.

Mutations in MIC26 were reported to result in mitochondrial myopathy, lactic acidosis, and cognition defects (Beninca et al, 2021), as well as a lethal progeria-like phenotype (Peifer-Weiß et al, 2023). Interestingly, there is an intricate connection between MIC26 and metabolic disorders. Patients with diabetes (Lamant et al, 2006) and dogs fed with a HFD for 9 wk (Philip-Couderc et al, 2003) showed increased Mic26 transcripts in the heart. Accordingly, adenovirus-mediated human MIC26 over-expression in mice, administered through the tail vein, led to increased levels of triacylglycerides (TAG) in murine plasma, when fed with HFD, and TAG accumulation in the murine liver (Tian et al, 2017). In another study, MIC26 transgenic mice hearts displayed an increase in diacylglycerides (DAG) but not TAG (Turkieh et al, 2014) as in the previous described study (Tian et al, 2017) suggesting modulatory roles of MIC26 in lipid metabolism. Recently, in mitochondria-rich brown adipose tissue (BAT), down-regulation of Mic26 mRNA and protein levels were reported in diet-induced or leptin-deficient obese (ob/ob) murine models compared with the respective controls. Mice with an adipose tissue-specific deletion of Mic26 which were fed with a HFD gained more total body weight and adipose tissue fat mass than control mice (Guo et al, 2023). Hence, we hypothesize that MIC26 has an unidentified regulatory role under nutrient-enriched conditions. Therefore, to understand the function of MIC26, we used WT and MIC26 KO cells as a model system under standard glucose culture conditions and excessive glucose culture conditions termed normoglycemia and hyperglycemia, respectively. We used a multi-omics approach encompassing transcriptomics, proteomics and targeted metabolomics to investigate the pathways regulated by MIC26. We found that the function of MIC26 is critical in various pathways regulating fatty acid synthesis, oxidation, cholesterol metabolism and glycolysis. Interestingly, we observed an entirely antagonistic effect of cellular de novo lipogenesis in MIC26 KO cells compared with WT cells depending on the applied nutrient conditions. This showed that the response to high glucose conditions is strongly dependent on the presence of MIC26. In addition, we found that cells deleted for MIC26 displayed alterations of mitochondrial glutamine usage and oxidative phosphorylation. Overall, we propose that MIC26 is a unique mitochondrial apolipoprotein functioning as a mitochondrial fuel sensor that regulates central metabolic pathways to meet mitochondrial and, thus, cellular energy demands.

# Results

### Mitochondrial apolipoproteins, MIC26, MIC27, and MIC25 are increased in cells exposed to hyperglycemia

There is a strong link between metabolic abnormalities and pattern of MIC26 expression. Increased levels of MIC26 transcripts were observed in diabetic patients (Lamant et al, 2006) and increased accumulation of lipids were found upon Mic26 overexpression in the mouse (Turkieh et al, 2014; Tian et al, 2017). To understand the role of MIC26 in cellular metabolism, we used hepatocyte-derived HepG2 cells as the cellular model and generated MIC26 KO cells using the CRISPR-Cas9 system. WT and MIC26 KO cells were grown in standard (5.5 mM) and excessive concentrations of glucose (25 mM), defined as normoglycemic and hyperglycemic conditions, respectively, throughout the manuscript, for a prolonged period of 3 wk to investigate long term effects of nutritional overload. Initially, we checked whether there is a difference in the amounts of various MICOS proteins in WT HepG2 cells grown in normoglycemia and hyperglycemia. Western blot (WB) analysis showed a significant increase in MIC26 and MIC27 along with MIC25 in cells grown in hyperglycemia compared with normoglycemia (Fig 1A and B). We did not observe any significant changes in the amounts of MIC19, MIC60, MIC10, and MIC13 in WT-hyperglycemia (WT-H) compared with WT-Normoglycemia (WT-N) condition. This pointed to a possible role of the MICOS subunits, MIC26, MIC27, and MIC25 in hyperglycemia compared with normoglycemia (Fig 1A and B).

The MICOS proteins regulate the IM remodelling by working in unison to maintain CJs and contact sites between IM and OM (Anand et al, 2021). Still, deficiency of different MICOS proteins shows variable effects on the extent of CJs loss and cristae ultrastructure (Weber et al, 2013; Kondadi et al, 2020a; Anand et al, 2020; Stephan et al, 2020). MIC10 and MIC60 have been considered as core regulators of IM remodelling displaying severe loss of CJs (Kondadi et al, 2020a; Stephan et al, 2020). The extent of mitochondrial ultrastructural abnormalities upon MIC26 deletion varies among different cell lines tested (Koob et al, 2015; Anand et al, 2020; Stephan et al, 2020). Therefore, we performed transmission electron microscopy (TEM) in WT and MIC26 KO HepG2 cells which revealed a reduction in cristae content (cristae number per unit mitochondrial length per mitochondria) in MIC26 KOs compared with WT cells in both nutrient conditions (Fig 1C and E). Thus, the loss of cristae was dependent on MIC26 and independent of the glucose concentration used in cell culture. In addition, there was a decrease in cristae number in WT cells grown in hyperglycemia compared with normoglycemia showing that higher glucose levels lead to decreased cristae density, which is a common phenotype in diabetic mice models (Bugger et al, 2008; Xiang et al, 2020). As the number of cristae were already decreased in certain conditions, we analyzed the number of CJs normalized to cristae number and found that a significant decrease in CJs was observed in MIC26 KOs independent of the nutrient conditions (Fig 1D and E). Overall, the loss of MIC26 leads to mitochondrial ultrastructural abnormalities accompanied by reduced number of cristae and CJs compared with WT cells (Fig 1C–E).

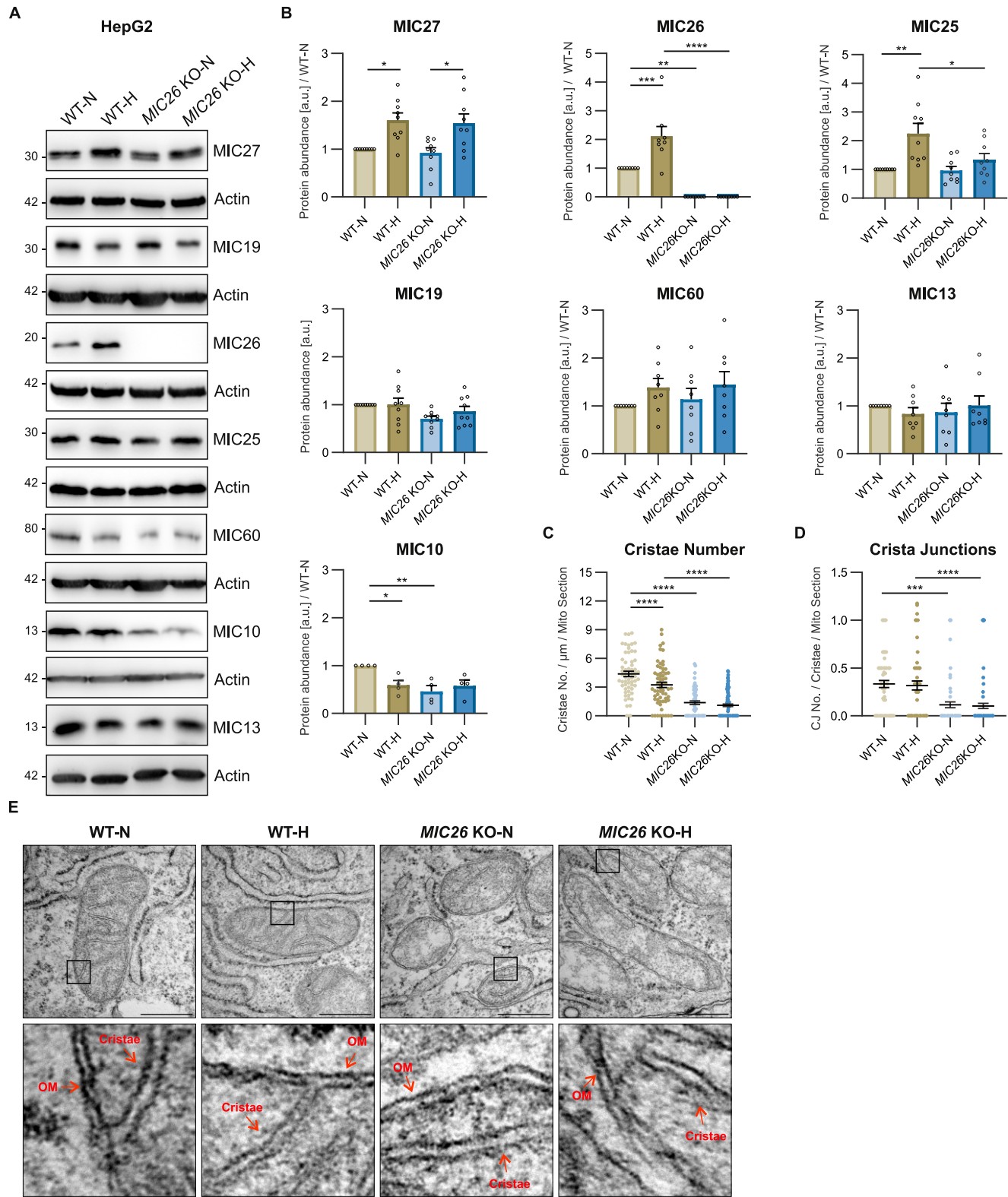

**Figure 1. Mitochondrial apolipoproteins MIC26 and MIC27 are selectively increased along with MIC25 in cells exposed to hyperglycemia.**
(A, B) Western blot analysis of all mitochondrial contact site and cristae organising system subunits from HepG2 WT and *MIC26* KO cells cultured in normo- and hyperglycemia (N = 4–8). Chronic hyperglycemia treatment leads to increased levels of MIC27, MIC26, and MIC25 in WT cells. Loss of MIC26 is accompanied by decreased MIC10 in normoglycemia. (C, D, E) Electron microscopy data including quantification of cristae number per unit length (µm) per mitochondrial section (C) and crista junctions per cristae per mitochondrial section (D), along with representative images (E) from HepG2 WT and *MIC26* KO cells cultured in normo- and hyperglycemia (N =

## Hyperglycemia confers antagonistic regulation of lipid and cholesterol pathways, in *MIC26* KO versus WT cells, compared with normoglycemia

To understand the role of MIC26 in an unbiased manner, we compared WT and *MIC26* KO cells cultured under respective nutrient conditions by using quantitative transcriptomics and proteomics analyses. A total of 21,490 genes were obtained after the initial mapping of the RNA-Seq data, of which 2,933 were significantly altered in normoglycemic *MIC26* KO compared with WT cells (fold change of ±1.5 and Bonferroni correction $P \leq 0.05$), whereas in hyperglycemia, *MIC26* KO had 3,089 significantly differentially expressed genes as compared with WT-N cells. A clustering analysis of identified transcripts involving all four conditions along with respective replicates is depicted (Fig S1A). A Treemap representation shows comparison of significantly up-regulated and down-regulated clustered pathways in *MIC26* KOs cultured in normoglycemia (Fig S1B and C) and hyperglycemia (Fig S1D and E) compared with WT. Interestingly, the pathways relating to sterol, cholesterol biosynthetic processes and regulation of lipid metabolic processes were significantly up-regulated in *MIC26* KO-N compared with WT-N. On the contrary, in *MIC26* KO-H compared with WT-H, pathways involved in sterol, secondary alcohol biosynthetic processes along with cholesterol biosynthesis and cellular amino acid catabolic processes including fatty acid oxidation (FAO) were mainly down-regulated. Thus, an antagonistic regulation is observed upon *MIC26* deletion when normoglycemia and hyperglycemia are compared. A detailed pathway enrichment analysis for significantly up-regulated genes in *MIC26* KO versus WT cells grown in normoglycemia also revealed genes involved in cholesterol, steroid biosynthetic pathways, fatty acid synthesis, and oxidation as well as glycolysis and gluconeogenesis (Fig 2A). The genes involved in cholesterol biosynthetic pathways, glycolysis and gluconeogenesis, FAO and fatty acid synthesis were significantly down-regulated in *MIC26* KOs grown in hyperglycemia compared with WT cells (Fig 2D). Conversely, there was no specific trend upon considering key metabolic pathways, when genes down-regulated in normoglycemia (Fig 2B) were compared with genes up-regulated in hyperglycemia (Fig 2C). Furthermore, we also observed an antagonistic behavior of cholesterol metabolism using the transcriptomics data (Fig 2A and D). Detailed analysis of the transcriptomics data in *MIC26* KO-N compared with WT-N showed that ≈80% of the genes regulating cholesterol biosynthesis were significantly up-regulated upon normoglycemia (Fig S2A), whereas the opposite was true for hyperglycemia (Fig S2B). At the proteome level, the effect of *MIC26* deletion was mainly observed in normoglycemic conditions where 8 of 12 detected proteins involved in cholesterol biosynthesis showed a significant increase in peptide abundances, whereas this increase was diminished in *MIC26* KO-H compared with WT-H cells (Fig S2C–N). Thus, the loss of MIC26

strongly impacts cholesterol biosynthesis in a nutrient-dependent manner. In conclusion, under normoglycemic conditions, MIC26 acts as a repressor of cholesterol biosynthesis whereas under hyperglycemic conditions MIC26 rather drives this pathway. We further used targeted metabolomics to decipher any altered cholesterol biosynthesis by quantifying the cholesterol amounts at steady state. In accordance with cholesterol synthesis promoting role of MIC26 in hyperglycemia, cholesterol levels were strongly reduced in *MIC26* KO-H compared with WT-H cells. Moreover, cholesterol levels were significantly increased in WT-H cells compared with WT-N which was not the case and even reversed in *MIC26* KO cells (Fig S2O). Thus, MIC26 is required to maintain cholesterol homeostasis and cellular cholesterol demand in a nutrient-dependent manner and is of particular importance under hyperglycemia.

## MIC26 maintains the glycolytic function

Besides an antagonistic regulation of the cholesterol biosynthetic pathway, we also observed an opposing trend of genes involved in lipid metabolism and glycolysis (indicated using arrows in Fig 2A and D). To gain further insights about the role of MIC26 regarding the differential regulation of glycolytic pathways in normoglycemia and hyperglycemia, we re-visited our transcriptomics (Fig S3) and proteomics (Figs 3A–C and J–L and S3C) datasets and investigated the genes regulating glycolysis upon *MIC26* deletion. On the one hand, in *MIC26* KO-N compared with WT-N, we found that the transcripts encoding hexokinase (*HK*) 1, phosphofructokinase 1 (*PFKP*) (Fig S3A), and aldolase (ALDOA & ALDOC), phosphoglycerate kinase (PGK1), pyruvate kinase (PKM & PKLR) protein levels were significantly up-regulated (Figs 3A and S3C), whereas glyceraldehyde-3-phosphate dehydrogenase (*GAPDH*) (Fig 3B) and enolase (*ENO*2) were down-regulated (Fig S3A). On the other hand, in *MIC26* KO-H compared with WT-H, we observed decreased GAPDH and glucose-6-phosphate isomerase (GPI) proteins and transcripts (Figs 3B and C and S3B). This could indicate that in hyperglycemia, deletion of *MIC26* leads to deregulation of the glycolysis pathway resulting in increased accumulation of glucose (Fig 3D) and decreased glycolysis end products. Therefore, to evaluate the metabolic effect of differentially expressed genes (Fig S3A and B) and proteins involved in glucose uptake (Fig 3J–L) and glycolysis (Figs 3A–C and S3C), we checked whether the glycolytic function is altered in *MIC26* KOs using a Seahorse Flux Analyzer with the glycolysis stress test (Fig 3E–H). Based on the extracellular acidification rate (ECAR), the "glycolytic reserve" is an index of the ability to undergo a metabolic switch to glycolysis achieved by the cells upon inhibition of mitochondrial ATP generation whereas the "glycolytic capacity" measures the maximum rates of glycolysis which the cell is capable to undergo. Overall "glycolytic function" is measured after cellular glucose deprivation for 1 h and subsequently by quantifying the

2). Loss of MIC26 led to decreased cristae number and crista junctions independent of normo- and hyperglycemia. Red arrows in lower row indicate outer membrane (OM) or cristae. Scale bar represents 500 nm. **(B, C, D)** Data are represented as mean ± SEM. Statistical analysis was performed using one-way ANOVA with *$P < 0.05$, **$P < 0.01$, ***$P < 0.001$, ****$P < 0.0001$ for all meaningful combinations (except comparison of WT-H with *MIC26* KO-N and vice versa). Non-significant *P*-values are not shown. N represents the number of biological replicates.

Source data are available for this figure.

**A**

### Normoglycemia *MIC26* KO vs. WT
### Upregulated Wiki Pathways

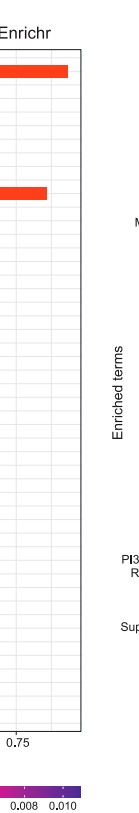

**B**

### Normoglycemia *MIC26* KO vs. WT
### Downregulated Wiki Pathways

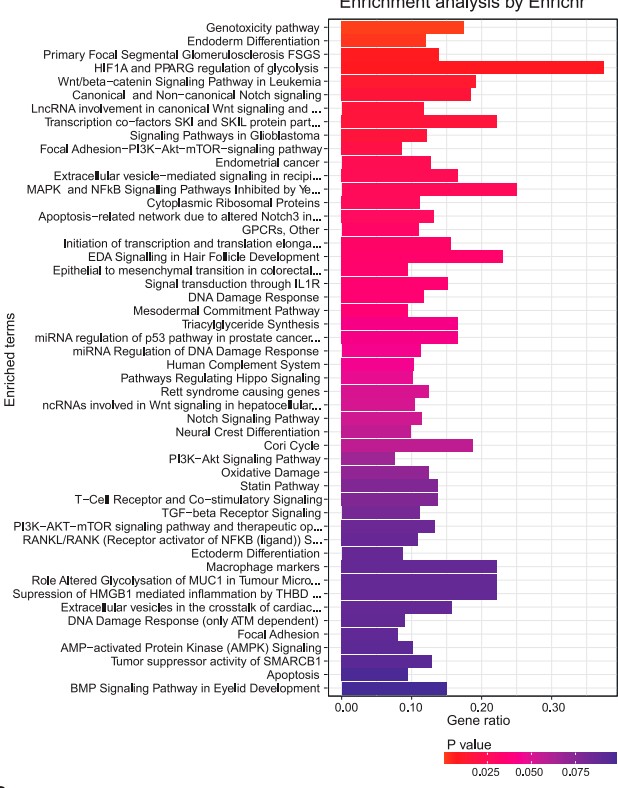

**C**

### Hyperglycemia *MIC26* KO vs. WT
### Upregulated Wiki Pathways

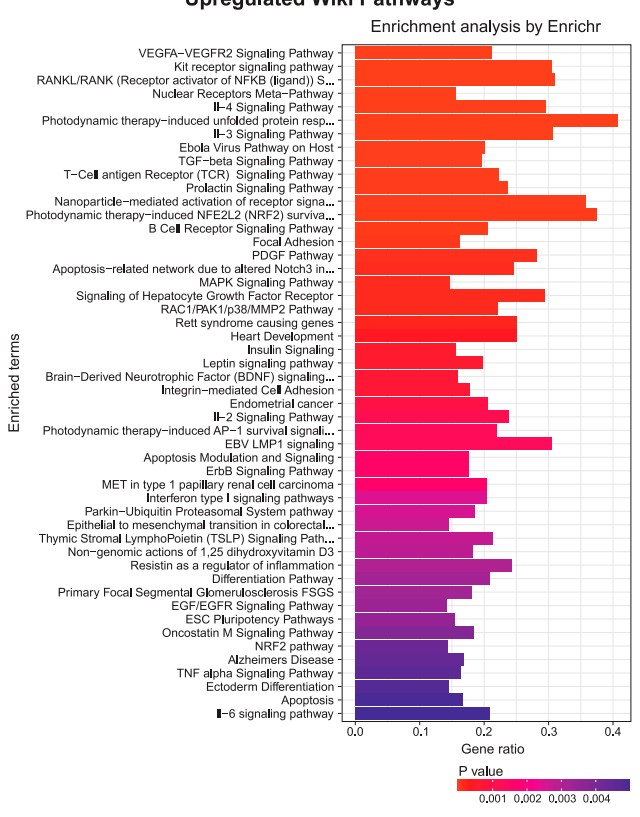

**D**

### Hyperglycemia *MIC26* KO vs. WT
### Downregulated Wiki Pathways

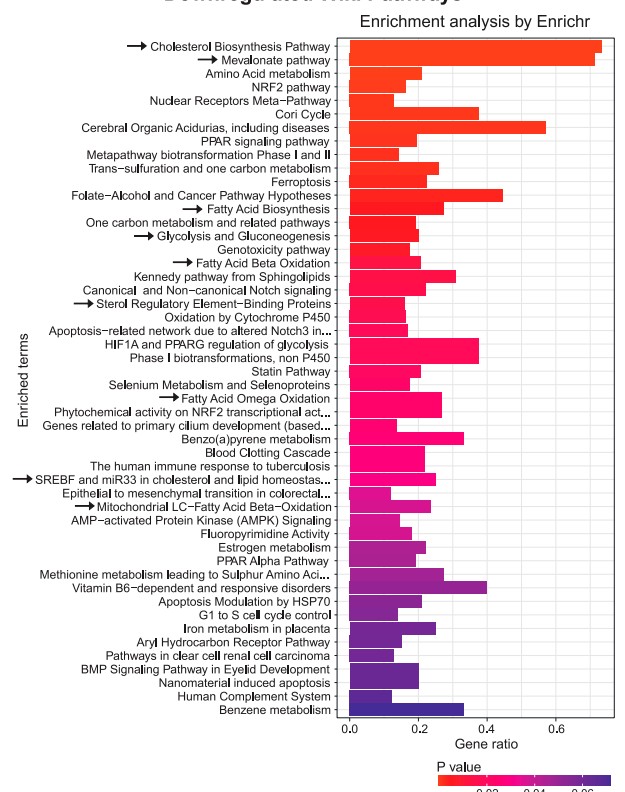

ECAR primarily arising from cellular lactate formation after providing the cell with saturating glucose amounts. We observed that the glycolytic reserve was significantly increased only in cells cultured in normoglycemia and not in hyperglycemia upon deletion of *MIC26* (Fig 3F), whereas the glycolysis function and glycolytic capacity were not significantly increased in *MIC26* KO under both nutrient conditions (Fig 3G and H). Therefore, the ability of *MIC26* KO cells (compared with WT) to respond to energetic demand by boosting glycolysis is increased under normoglycemia, whereas *MIC26* KO cells primed to hyperglycemia were not able to increase glycolytic reserve indicating a clearly different regulation of glycolysis under normoglycemia versus hyperglycemia. Because the MICOS complex consists of two subcomplexes: MIC19/25/60 and MIC10/26/27/13, we wanted to understand if the role of MIC26 in maintaining glycolytic function is specific for MIC26 or if the observed alterations also occur in other MICOS KO cell lines. In this endeavour, we used MICOS knockouts from the two distinct subcomplexes, *MIC19* and *MIC27* KO HepG2 cells, respectively, and cultured them in normoglycemic or hyperglycemic conditions for 3 wk similar to *MIC26* KOs (Fig S4) and measured the various parameters of glycolytic function (Fig S5). We compared the glycolytic reserve (Fig S5B), function (Fig S5C), and capacity (Fig S5D) in *MIC27* and *MIC19* KOs with *MIC26* KOs (Fig 3F–H), respectively. The previously observed twofold increase in glycolytic reserve in *MIC26* KO cells (Fig 3F) was absent in *MIC27* and *MIC19* KO cells (Fig S5B) when compared with respective control cells. This indicates a specific role of MIC26 in regulating the glycolysis while excluding the specificity of both the MICOS subcomplexes.

To understand the reason for increased glycolytic reserve in *MIC26* KO cells, we quantified the intracellular glucose levels, at steady state in WT and *MIC26* KO, which were significantly increased upon *MIC26* deletion only in hyperglycemia but not normoglycemia compared with the respective WT cells (Fig 3D). We further checked whether the increased glucose levels in cells cultured in hyperglycemia is because of increased glucose uptake. In normoglycemia, a glucose uptake assay showed a modest but significant increase in glucose uptake in *MIC26* KOs compared in WT cells (Fig 3I) which is consistent with a strong increase in GLUT3 amounts (Fig 3J) albeit accompanied by a down-regulation of GLUT1 upon *MIC26* depletion (Fig 3K). GLUT2 levels remained unchanged in all conditions (Fig 3L). However, the observed increased glucose uptake in *MIC26* KO-N (compared with WT-N) was abolished in *MIC26* KO-H (compared with WT-H) and accordingly accompanied by no increase in GLUT3 levels showing that the high amounts of glucose in *MIC26* KO cells grown in hyperglycemia cannot be explained by an increased glucose uptake under these conditions (Fig 3I). In *MIC26* KO-N compared with WT-N, even though we observed an increase in glucose uptake, the amount of glycolysis end products, namely pyruvate (Fig 3M) and lactate (Fig 3N) were unchanged. In hyperglycemia, a significant reduction in pyruvate (Fig 3M) and lactate

(Fig 3N) amounts were observed at steady-state despite increased glucose levels upon *MIC26* deletion. Overall, upon *MIC26* deletion pyruvate and lactate levels were decreased in hyperglycemia, whereas no change was observed in normoglycemia. These results combined with the already discussed differentially regulated transcripts and proteins involved in glycolysis prompted us to check whether there is a difference of shuttling metabolic intermediates from glycolysis towards lipid anabolism. Glycerol-3-phosphate (G3P) is a precursor for lipid biosynthesis synthesized from dihydroxyacetone phosphate which is derived from glycolysis. We observed an increase in G3P levels upon *MIC26* deletion in normoglycemic conditions, whereas G3P levels were significantly reduced in *MIC26* KO-H cells, compared with the respective WT cells (Fig 3O). This opposing trend, together with the previously described antagonistic enrichment in fatty acid biosynthesis (Fig 2A and D), indicates that *MIC26* deletion rewires glycolytic function to drive lipogenesis in normoglycemia with an antagonistic effect in hyperglycemia. Furthermore, we checked the cellular effect of *MIC26* loss on lipid anabolism in normo- and hyperglycemia.

### The loss of MIC26 leads to metabolic rewiring of cellular lipid metabolism via CPT1 and dysregulation of fatty acid synthesis

The respective increase and decrease in G3P (Fig 3O) in normoglycemia and hyperglycemia upon *MIC26* deletion when compared with WT as well as an opposing trend in fatty acid biosynthesis reflected in our transcriptomics data (Fig 2A and D) prompted us to explore the regulation of cellular lipid metabolism. Lipid droplets (LDs) play a key role in energy metabolism and membrane biology by acting as reservoirs to store TAG and sterol esters which are released to the relevant pathways according to cellular demand (Thiam et al, 2013). Using BODIPY staining, we checked the cellular LD content in unstimulated and palmitate-stimulated WT and *MIC26* KO cells grown in normoglycemia and hyperglycemia, respectively. The number of LDs and the respective fluorescence intensity of BODIPY are indicative of cellular lipid content (Chen et al, 2022). We observed a general increase in LD number in *MIC26* KOs irrespective of palmitate treatment or not (Fig 4A and B). At steady state, there was no increase in LD number in *MIC27* KOs unlike *MIC26* KOs although we observed an increase in LD number upon palmitate treatment (Fig S6A, B, and D). In *MIC19* KOs, at steady-state, there was similar increase in LD number as compared with *MIC26* KOs. Therefore, overall, the regulation of LD number in *MIC26* KOs is similar to *MIC19* KOs but different to *MIC27* KOs. The increased intensity of BODIPY staining observed in normoglycemia was not evident in *MIC26* KO-H compared with respective WT cells (Fig 4C and D). Furthermore, when we fed free fatty acids (FFAs) in the form of palmitate, there was a further increase in BODIPY intensity in *MIC26* KOs in normoglycemia. In contrast, under hyperglycemia

---

**Figure 2. Hyperglycemia confers antagonistic regulation of lipid and cholesterol pathways, in *MIC26* KO versus WT cells, compared with normoglycemia.**
**(A, B, C, D)** WikiPathway enrichment using EnrichR analysis of differentially up-regulated and down-regulated genes (A, B) in normoglycemic *MIC26* KO and (C, D) in hyperglycemic *MIC26* KO cells compared with respective WT, respectively. Arrows indicate antagonistically regulated metabolic pathways including glycolysis, cholesterol biosynthesis, fatty acid synthesis, and oxidation. Differentially expressed genes were considered statistically significant with a cut-off fold change of ±1.5 and Bonferroni correction *P* ≤ 0.05.

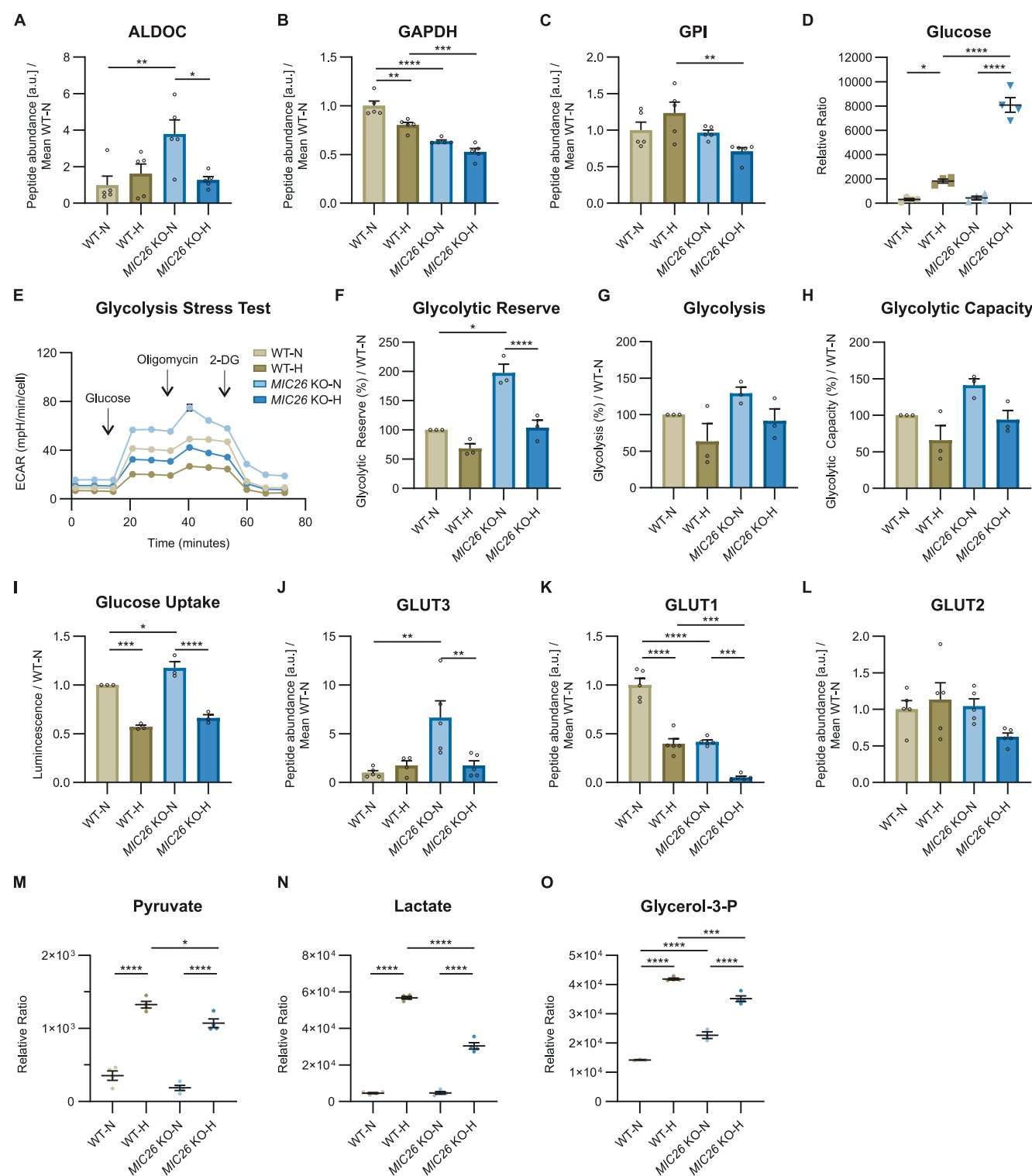

**Figure 3. MIC26 maintains the glycolytic function.**
**(A, B, C)** Peptide abundances of enzymes involved in glycolysis pathway curated from proteomics data (N = 5). **(D)** Steady-state metabolomics (GC-MS) data reveal increased cellular glucose accumulation upon *MIC26* deletion in hyperglycemia (N = 3–4). **(E)** Representative glycolysis stress test seahorse assay analysis, with sequential injection of glucose, oligomycin, and 2-deoxyglucose, reveals a tendency towards increased glycolysis upon *MIC26* deletion (n = 23). **(F, G, H)** Quantification from various biological replicates shows a significant increase in cellular glycolytic reserve in normoglycemic, but not in hyperglycemic conditions (F). **(G, H)** A non-significant tendency towards increased glycolysis (G) and glycolytic capacity (H) was observed upon *MIC26* KO in normoglycemic, but not in hyperglycemic conditions (N = 3). **(I)** Cellular glucose uptake was measured using Glucose Uptake-Glo assay normalized to WT-N. *MIC26* deletion leads to an increased glucose uptake upon normoglycemia (N = 3). **(J, K, L)** Peptide abundances of transporters involved in glucose uptake namely GLUT3 (J), GLUT1 (K), and GLUT2 (L) curated from proteomics data (N = 5). **(M, N)**

*MIC26* KO cells showed lower LD intensity when compared with WT cells again demonstrating an antagonistic role of MIC26 when normoglycemia was compared with hyperglycemia (Fig 4C and D). This antagonistic regulation of the intensity of BODIPY stain was not observed in *MIC19* KO cells as there were no differences when compared with control cells either with or without palmitate treatment (Fig S6A, C, and D). Overall, taking the LD number and fluorescence intensity of the BODIPY staining into account, we conclude that MIC26 has a differential regulation compared with both MIC27 and MIC19. These experiments allow us to conclude that the specific effect of *MIC26* deletion on LD accumulation depends on the nutrient condition which is enhanced under nutrient-rich (high-glucose/high-fat-like) conditions. Overall, MIC26 is essential to regulate the amount of cellular LD content in a nutrient-dependent manner (Fig 4D).

LD biogenesis is closely linked to increased cellular FFA levels (Zadoorian et al, 2023). Using targeted metabolomics, we investigated the steady-state levels of long-chain FFAs in WT and *MIC26* KO cell lines cultured in normoglycemia and hyperglycemia (Fig 4E). We identified that there was either no change or an increase in saturated FFAs including lauric (12:0), myristic (14:0), palmitic (16:0), stearic (18:0), arachidic (20:0), and behenic (22:0) acid in normoglycemia in *MIC26* KOs compared with WT cells (Figs 4E and S7). In contrast, we consistently found a decrease in most of the above-mentioned saturated FFAs in *MIC26* KO-H compared with WT-H. This trend was also observed in unsaturated FFAs such as oleic acid (18:1). Overall, we conclude that there is a consistent decrease in saturated FFAs in *MIC26* KO-H as opposed to *MIC26* KOs grown in normoglycemic conditions consistent to the observed trend in LD formation. Increased level of FFAs and LDs can arise from increased FFAs biosynthesis and reduced FFA catabolism via mitochondrial *β*-oxidation (Afshinnia et al, 2018). Mitochondrial *β*-oxidation requires import of long-chain FFAs using the carnitine shuttle comprised of carnitine palmitoyl transferase 1 (CPT1) and 2 (CPT2) and carnitine-acylcarnitine translocase (CACT), into the mitochondrial matrix. Depletion of CPT1A, which is the rate limiting step of FAO, coincides with lipid accumulation in the liver (Sun et al, 2021). Therefore, we determined the CPT1A amounts using WB analysis (Fig 4F and G) which were in line with transcriptomics and quantitative PCR data (Fig S8A and B). *MIC26* deletion revealed a reduction in CPT1A in normoglycemia compared with WT cells. In WT cells, hyperglycemia already triggered a reduction in CPT1A level and there was no further decrease in CPT1A in *MIC26* KOs grown in hyperglycemia (Fig 4F and G). To understand the functional significance of CPT1A reduction on mitochondrial function, we checked the FAO capacity of respective cell lines by feeding them with palmitate and analyzing the induced basal respiration and spare respiratory capacity (SRC) of mitochondria compared with BSA control group (Figs 4H and I and S8C and D). SRC is the difference between FCCP stimulated maximal respiration and basal oxygen consumption and therefore

is the ability of the cell to respond to an increase in energy demand. We observed a significant reduction in palmitate-induced basal respiration and SRC in *MIC26* KO-N compared with WT-N determining decreased mitochondrial long-chain fatty acid *β*-oxidation. It is important to note that we already observed a significant decrease in mitochondrial *β*-oxidation in WT-H condition which was not further affected in *MIC26* KO-H in agreement with the reduced CPT1A levels (Fig 4F and G). We further analyzed the reduction in oxygen consumption rate (OCR) induced by etomoxir inhibition of CPT1A (Fig 4J). In *MIC26* KO-N compared with WT-N, palmitate-induced OCR was reduced moderately, yet this was not significant. For the respective hyperglycemic conditions, we did not observe a change which was again in line with the observed CPT1A levels. Thus, reduced *β*-oxidation in *MIC26* KO-N compared with WT-N is apparently contributing to increased FFA levels and LD content and could be mediated, at least in part, via the reduced levels of CPT1A resulting in reduced transport of FFAs into mitochondria.

We further checked whether FFA biosynthesis plays a role in the nutrition-dependent antagonistic regulation of lipid anabolism in *MIC26* KO cell line. FFA biosynthesis is initiated with the export of citrate generated in TCA cycle from mitochondria to the cytosol. The export is mediated by the citrate/malate exchanger SLC25A1 which is present in the mitochondrial IM. Proteomics and transcriptomics data showed that SLC25A1 was increased in normoglycemia in *MIC26* KOs (compared with respective WT) but not in hyperglycemia (Fig S8E and F). We then checked for further changes in the transcriptome and proteome levels of key enzymes playing a role in FFA synthesis. We found that ATP citrate lyase (ACLY, Fig S8G), acetyl-Co-A carboxylase (ACACA, Fig S8H and I) which converts acetyl-CoA into malonyl-CoA, fatty acid synthase (FASN, Fig S8J and K) and acetyl-CoA desaturase (SCD, Fig S8L and M) were increased in normoglycemia in *MIC26* KOs but mostly unchanged in hyperglycemia. In addition, hyperglycemia resulted in an increase in glycerol kinase (GK) in WT cells which was absent in *MIC26* KO cells (Fig S8N). Therefore, our data indicate that the FFA biosynthesis pathway is up-regulated upon loss of *MIC26* KO in normoglycemia but not in hyperglycemia compared with respective WT conditions. An up-regulation of FFA biosynthesis along with reduced mitochondrial *β*-oxidation partially mediated by reduced CPT1A amount in *MIC26* KO-N and a shift of glycolytic intermediates resulting in G3P accumulation show that loss of MIC26 leads to a cumulative metabolic rewiring towards increased cellular lipogenesis.

### *MIC26* deletion leads to hyperglycemia-induced decrease in TCA cycle intermediates

To synthesize FFA, citrate first needs to be generated by the TCA cycle in the mitochondrial matrix before it is exported to the cytosol. Using targeted metabolomics, we checked whether the TCA

---

Steady-state metabolomics (GC-MS) shows unaltered cellular pyruvate (M) and lactate (N) levels in *MIC26* KO cell lines in normoglycemia but decreased levels upon *MIC26* deletion in hyperglycemia (N = 3–4). **(O)** *MIC26* deletion increases glycerol-3-phosphate amount in normoglycemia with an antagonistic effect in hyperglycemia compared with the respective WT (N = 3–4). **(A, B, C, D, E, F, G, H, I, J, K, L, M, N, O)** Data are represented as mean ± SEM. Statistical analysis was performed using one-way ANOVA with *P < 0.05, **P < 0.01, ***P < 0.001, ****P < 0.0001 for all meaningful combinations (except comparison of WT-H with *MIC26* KO-N and vice versa). Non-significant *P*-values are not shown. N represents the number of biological replicates and n the number of technical replicates.

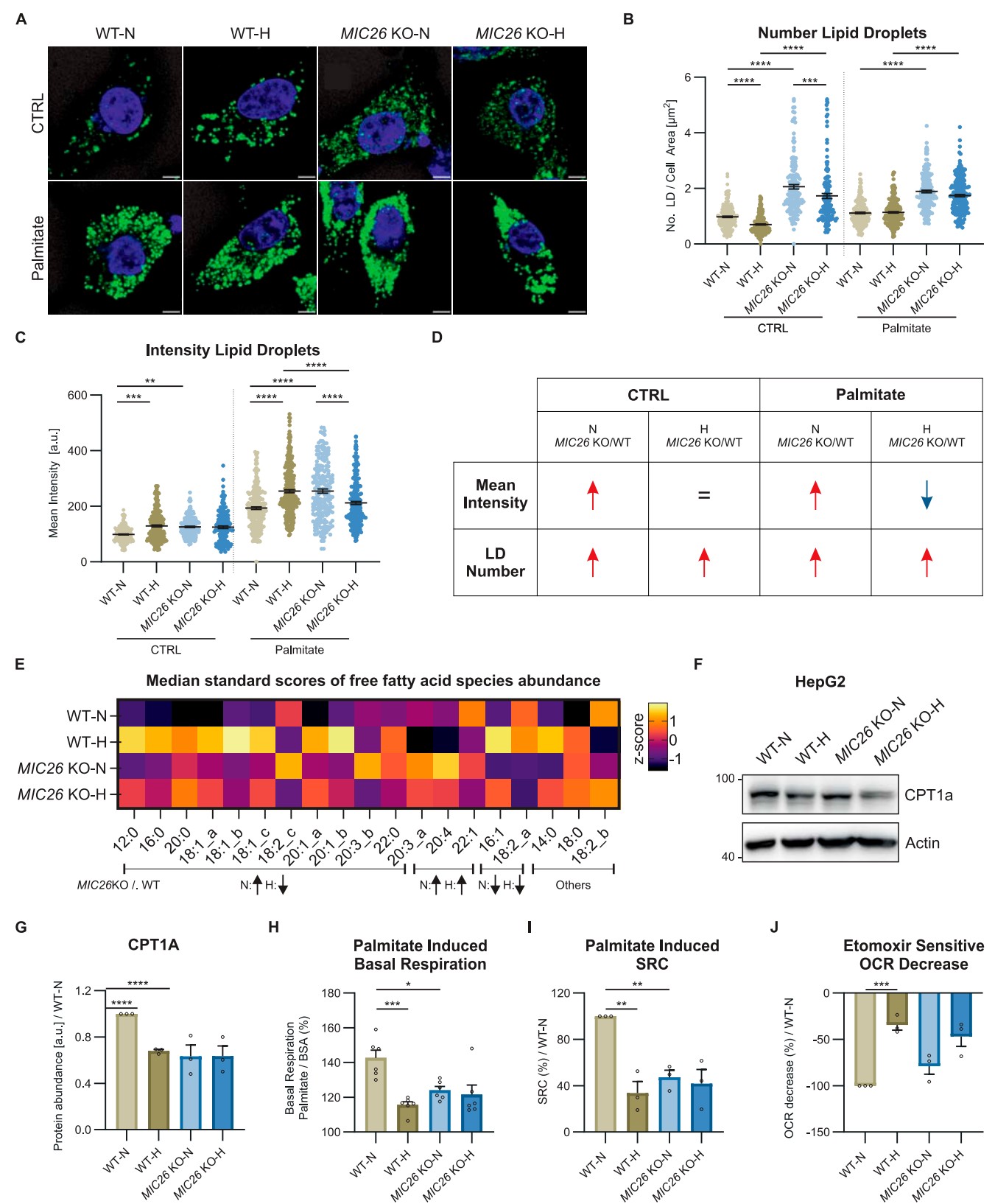

**Figure 4. The loss of MIC26 leads to metabolic rewiring of cellular lipid metabolism via CPT1A and dysregulation of fatty acid synthesis.**
**(A, B, C, D)** Analysis of lipid droplet formation in WT and *MIC26* KO cells cultured in normo- and hyperglycemia either in standard growth condition (CTRL) or upon palmitate stimulation (100 μM, 24 h). **(A)** Representative confocal images of lipid droplets stained using BODIPY 493/503 are shown (A). **(B, C)** Quantification shows number

cycle metabolism is altered upon *MIC26* deletion at steady state in both nutrient conditions (Fig 5A). As previously described, glycolysis resulted in decreased pyruvate levels upon *MIC26* deletion in hyperglycemia, whereas no change was observed in normoglycemia (Fig 3M). Furthermore, most of the downstream metabolites including (iso-)citrate, succinate, fumarate and malate consistently showed a significant decrease in *MIC26* KO cells cultured in hyperglycemia, compared with WT condition, but not in normoglycemia following the previously observed trend in pyruvate levels. To elucidate a possible defect of mitochondrial pyruvate import, we checked mitochondrial pyruvate carrier 1 (MPC1) and MPC2 abundances (Fig 5B and C) as well as mitochondrial respiration after blocking mitochondrial pyruvate carrier (glucose/pyruvate dependency) using UK5099 inhibitor (Fig 5D). Whereas we observed a down-regulation of MPC1 in *MIC26* KO-N compared with WT-N, MPC1 abundances in *MIC26* KO-H compared with WT-H remained unchanged. Furthermore, we did not observe any changes in MPC2 level. Also, mitochondrial glucose/pyruvate dependency remained unchanged in the respective hyperglycemia combination, whereas we observed a minor but significant decrease in *MIC26* KO-N compared with WT-N. In addition, elucidation of abundances of mitochondrial enzymes catalyzing TCA cycle metabolites (Fig S9A–K) and the respective cytosolic enzymes (Fig S9L–N) interestingly revealed an up-regulation of citrate synthase (Fig S9A) and mitochondrial aconitase 2 (Fig S9B) in *MIC26* KO cells independent of nutrient conditions. Furthermore, the immediate downstream enzyme isocitrate dehydrogenase 2 which generates α-ketoglutarate (α-KG) was up-regulated in *MIC26* KO condition (Fig S9C). In contrast to all previously described metabolites, the α-KG levels were increased in hyperglycemia in *MIC26* KO compared with WT. The accumulation of α-KG possibly arises from a significant down-regulation of α-KG dehydrogenase in *MIC26* KO independent of the nutrient condition (Fig S9E). After that, an accumulation of α-KG by down-regulation of α-KG dehydrogenase would further explain the decreased formation of succinate in *MIC26* KO-H compared with WT-H. Succinate dehydrogenases (Fig S9G and H) and fumarase (Fig S9I) did not show any changes in abundances upon the respective *MIC26* KO to WT comparison reflecting the uniform metabolite trend in succinate, fumarate and malate. Overall, we observed a general decrease in several TCA cycle metabolites in *MIC26* KO-H compared with WT-H. Therefore, we propose that a down-regulation of FFA biosynthesis in *MIC26* KO-H compared with WT-H results from a limited formation of citrate via the mitochondrial TCA cycle presumably arising from reduced use of glucose.

## Aberrant glutamine metabolism is observed in *MIC26* KOs independent of nutritional status

Glutaminolysis feeds α-KG in the TCA cycle. To check whether the increase in α-KG could be (apart from down-regulation of α-KG dehydrogenase amounts) derived from glutaminolysis, we also checked glutamine (Fig 6A) and glutamate levels (Fig 5A). The amounts of glutamine at steady-state were uniformly increased in *MIC26* KOs irrespective of nutrient conditions (Fig 6A). Glutamate was decreased in *MIC26* KO-N compared with WT-N (Fig 5A). Mitochondria mainly oxidise three types of cellular fuels namely pyruvate (from glycolysis), glutamate (from glutaminolysis), and FFAs. We used a "mito fuel flex test" for determining the contribution of glutamine as a cellular fuel. The contribution of glutamine as cellular fuel could be determined using BPTES, an allosteric inhibitor of glutaminase (GLS1), which converts glutamine to glutamate. The extent of reduction in mitochondrial oxygen consumption upon BPTES inhibition is used as a measure for determining the glutamine dependency, whereas the capacity is the ability of mitochondria to oxidise glutamine when glycolysis and FFA oxidation are inhibited. Intriguingly, we observed that the *MIC26* KOs do not depend on glutamine as a fuel (Fig 6B, left histogram). However, they still can use glutamine when the other two pathways were inhibited (Fig 6B, right histogram). The glutamine oxidation capacity of *MIC26* KO cells cultured in normoglycemia and hyperglycemia appears slightly decreased compared with WT but this decrease is not statistically significant. Furthermore, to check if the ability of *MIC26* KO cells to bypass glutamine as a fuel is also observed in general for the MICOS complex proteins or only for one of the two MICOS subcomplexes, we used *MIC27* KO and *MIC19* KO cells to determine their dependency on glutamine oxidation (Fig S10F). We observed that there was no significant decrease in glutamine dependency in both *MIC27* and *MIC19* KOs. Furthermore, there was no difference in glutamine oxidation capacity when *MIC27* and *MIC19* KO cells were compared with control cells again ascertaining that MIC26 has an MICOS-independent role in regulating glutamine metabolism. Overall, we observe a remarkable metabolic rewiring of *MIC26* KOs to bypass glutaminolysis.

To understand whether the independency on glutamine as fuel arises because of the possibility of aberrant transport of glutamine into the mitochondria, we analyzed transcripts and proteins that were not only significantly down-regulated but also present in the mitochondria IM and interacted with MIC26. For this, we investigated putative MIC26 interactors by compiling a list using BioGRID, NeXtProt, and IntAct databases. SLC25A12, an antiporter of

---

of lipid droplets normalized to the total cell area ($\mu m^2$) (B) and mean fluorescence intensity per cell normalized to mean intensity of WT-N in all biological replicates (C). **(D)** *MIC26* deletion leads to a nutritional-independent increase in lipid droplet number (D). However, an opposing effect, leading to increase or decrease in mean fluorescence intensity of lipid droplets, upon comparison of *MIC26* KO to WT was observed in normo- and hyperglycemia respectively, with a pronounced effect upon feeding palmitate (N = 3). Scale bar represents 5 $\mu m$. **(E)** Heat map representing the abundance of steady state fed free fatty acid species in WT and *MIC26* KO cells cultured in normo- and hyperglycemia. 11 of 19 of the fed free fatty acid species represent an antagonistic behavior upon comparing *MIC26* KO to WT in normo- (increase) and hyperglycemia (decrease) (N = 3–4). **(F, G)** Western blot analysis (F), along with respective quantification (G) of WT and *MIC26* KO cells cultured in normo- and hyperglycemia, show a reduction in CPT1A in WT-H, *MIC26* KO-N and *MIC26* KO-H compared with WT-N (N = 3). **(H, I, J)** Mitochondrial fatty acid oxidation analyzed using Seahorse XF analyzer shows a decreased palmitate-induced basal respiration (H) and spare respiratory capacity (I) and a non-significant reduction in etomoxir-sensitive OCR decrease upon comparing *MIC26* KO to WT in normoglycemia (N = 3). **(B, C, G, H, I, J)** Data are represented as mean ± SEM. Statistical analysis was performed using one-way ANOVA with *$P < 0.05$, **$P < 0.01$, ***$P < 0.001$, ****$P < 0.0001$ for all meaningful combinations (except comparison of WT-H with *MIC26* KO-N and vice versa). Non-significant *P*-values are not shown. N represents the number of biological replicates.
Source data are available for this figure.

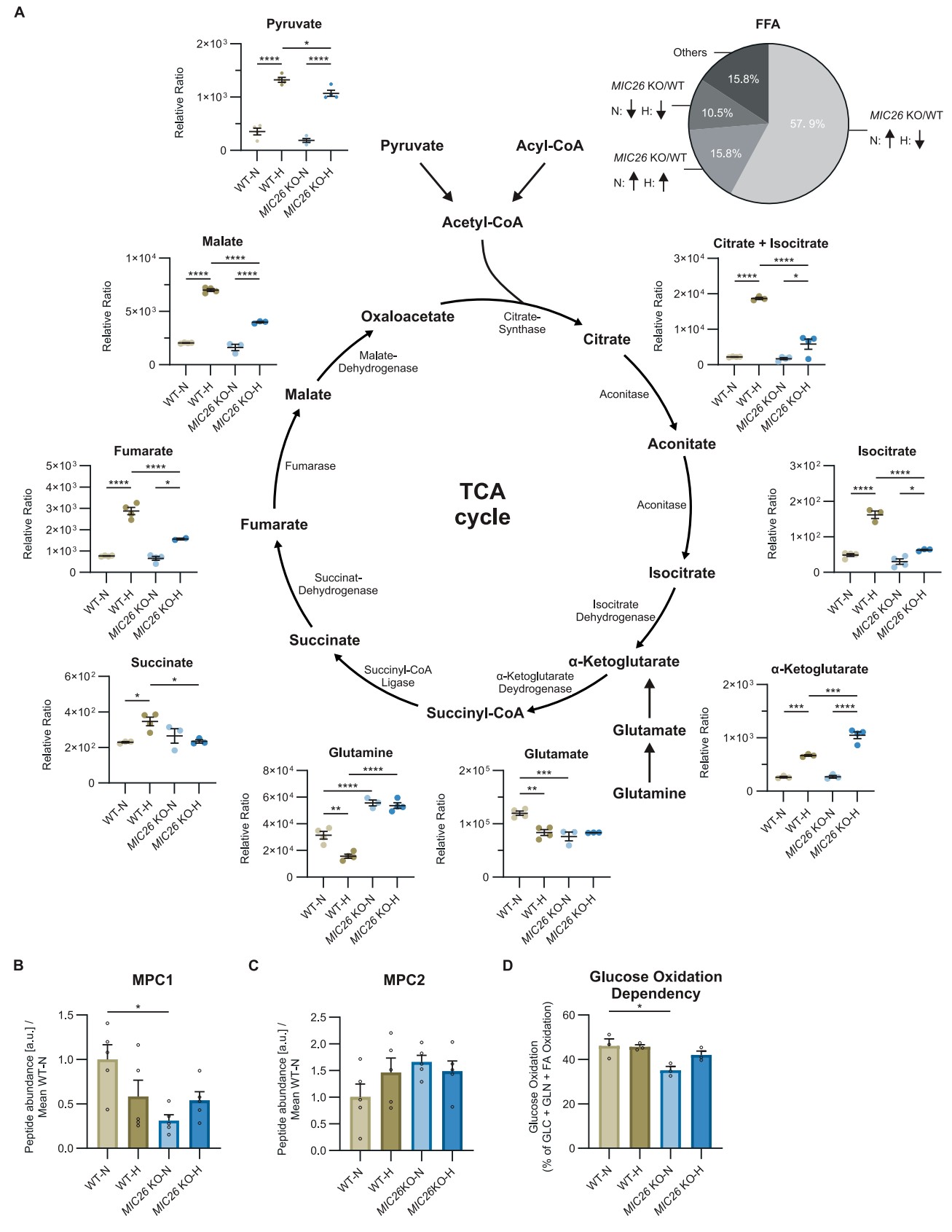

**A**

cytoplasmic glutamate and mitochondrial aspartate, was significantly down-regulated (Fig S10A) when showing up in the interactome of MIC26 (Fig S10E). Accordingly, WB analysis reveals a reduction in SLC25A12 in *MIC26* KOs compared with WT HepG2 cells in both normoglycemia and hyperglycemia (Fig 6C and D). Furthermore, it is known that a variant of SLC1A5 transcribed from an alternative transcription start site and present in the mitochondrial IM is responsible for transporting glutamine into mitochondria (Yoo et al, 2020a). Transcriptomics data revealed a reduction in SLC1A5 in *MIC26* KOs, whereas proteomics revealed a significant reduction in normoglycemia and non-significant reduction in hyperglycemia in *MIC26* KOs compared with WT (Fig S10B and C). An increase in cellular glutamine levels in *MIC26* KOs (Fig 6A) along with reduced levels of SLC1A5 and reduced mitochondrial glutamine dependency (Fig 6B) indicate a reduced transport of glutamine destined for glutaminolysis into mitochondria.

To delineate whether the increased glutamine levels at steady-state are because of decreased glutamine use or increased flux, we performed a metabolic tracing experiment where WT and *MIC26* KO cells, cultured in normoglycemia and hyperglycemia, were fed with labelled glutamine [U-$^{13}C_5$, $^{15}N_2$] for 0.5 and 6 h (Fig 6E–O). Glutamine is converted to glutamate by GLS in the mitochondria. The GLS amounts were not altered in *MIC26* KO cells when compared with respective WT cells grown in normoglycemia and hyperglycemia (Fig S10D). In line, label enriched glutamate species (m + 1 – m + 4) did not show major differences in all four conditions at both timepoints (Fig 6E). After this, we hypothesize that accumulation of glutamine in *MIC26* KO cells arises from other cellular pathways using glutamine being impaired, for example, synthesis of purine, pyrimidine, or amino acids. However, labelled α-KG (m + 1 – m + 5) was increased upon *MIC26* deletion with a pronounced effect in cells cultured in hyperglycemia similar to the detected steady-state amounts of α-KG (Fig 6F). To check the conversion rates of different metabolite reactions, we determined the enzyme conversion rates by calculating the ratio of the highest labelled species from the end-metabolite compared with the starting-metabolite. In accordance to the observed level of α-KG, the conversion ratio from glutamate to α-KG was significantly increased in *MIC26* KO cells (Fig 6K). We further checked the flux of TCA metabolites downstream to α-KG namely succinate, fumarate and malate. Despite the increased α-KG levels, the labelled succinate species (m + 1 – m + 4) was decreased in *MIC26* KO cells (Fig 6H). In line, the conversion rate from α-KG to succinate was significantly down-regulated in *MIC26* KO cells independent of glucose concentrations and timepoints (Fig 6M). However, the conversion ratio from succinate to fumarate catalyzed by mitochondrial complex II subunits, succinate dehydrogenases A-D, was increased in *MIC26* KO cell lines at the 6 h timepoint compared with WT in both normoglycemia and

hyperglycemia (Fig 6N). Despite the increase in fumarate conversion, the labelled fumarate and malate were decreased in *MIC26* KO compared with WT in normoglycemia but not in hyperglycemia (Fig 6I and J), whereas there were minor differences at 0.5 h. The conversion ratio from α-KG to malate was decreased upon *MIC26* deletion in both nutrient conditions at 0.5 and 6 h of glutamine labelling. Thus, despite increased conversion of succinate to fumarate and increased flux from glutamate to α-KG (in hyperglycemia) upon loss of MIC26, cellular glutaminolysis does not function optimally. We also checked the labelled citrate levels which showed minor changes after 0.5 h treatment but a major change in all labelled species (m + 1 – m + 5) after 6 h (Fig 6G). Correspondingly, the levels of citrate in WT-N cells were highly increased compared with all three other conditions. Conversion rates from α-KG (m + 5) to citrate (m + 5) were significantly reduced in *MIC26* KO cell lines compared with the respective WT cells (Fig 6L). Overall, the flux of glutamine through the TCA cycle is accompanied by decreased conversion of TCA cycle intermediates. Therefore, we conclude that aberrant glutaminolysis is observed upon loss of MIC26.

## MIC26 regulates mitochondrial bioenergetics by restricting the ETC activity and OXPHOS (super-)complex formation

We have shown that the loss of *MIC26* leads to dysregulation of various central fuel pathways. To understand the effect of *MIC26* deletion on cellular bioenergetics, we checked the mitochondrial membrane potential (ΔΨ$_m$) of WT and *MIC26* KO cells in both nutrient conditions by using TMRM dye (Fig 7A and B). Loss of MIC26 leads to decreased ΔΨ$_m$ compared with control cells in both normoglycemia and hyperglycemia. It is well known that mitochondrial loss of membrane potential is connected to mitochondrial dynamics (Giacomello et al, 2020). Thus, we checked the mitochondrial morphology and observed that loss of MIC26 consistently leads to a significant increase in mitochondrial fragmentation compared with WT-N (Fig 7C and D). In addition, WT cells grown in hyperglycemia despite maintaining the ΔΨ$_m$ exhibited fragmented mitochondria. We also checked the levels of major mitochondrial dynamic regulators: MFN1, MFN2, DRP1, and OPA1 processing into short forms. WB analysis showed that MFN1 levels were significantly decreased upon *MIC26* deletion in both normoglycemia and hyperglycemia compared with respective WT cells (Fig S11A and B) which could account for increased fragmentation. There was no major effect on the amounts of other factors which could account for mitochondrial fragmentation. Thus, *MIC26* deletion is characterized by reduced ΔΨ$_m$ and fragmentation of mitochondria which indicate altered mitochondrial bioenergetics. To determine this, we checked the mitochondrial function in WT and

**Figure 5. *MIC26* deletion leads to hyperglycemia-induced decrease in TCA cycle intermediates.**
**(A)** Representation of the relative amounts (GC-MS) of TCA cycle metabolites and associated precursors at steady state in WT and *MIC26* KO cells cultured in normo- and hyperglycemia. All the TCA cycle metabolites with the exception of α-ketoglutarate showed a decreasing trend upon *MIC26* KO when compared to WT in hyperglycemia (N = 3–4). **(B, C)** Mitochondrial pyruvate carrier 1 (MPC1) (B), but not MPC2 (C), is significantly decreased in *MIC26* KO-N compared to WT-N, as revealed by peptide abundances from proteomics data (N = 5). **(D)** Mitochondrial glucose/pyruvate dependency analysis, using Seahorse XF analyzer mito fuel flex test assay, reveals a decreased mitochondrial respiratory dependency of *MIC26* KO on glucose/pyruvate in normoglycemia (N = 3). **(A, B, C, D)** Data are represented as mean ± SEM. Statistical analysis was performed using one-way ANOVA with *$P < 0.05$, **$P < 0.01$, ***$P < 0.001$, ****$P < 0.0001$ for all meaningful combinations (except comparison of WT-H with *MIC26* KO-N and vice versa). Non-significant *P*-values are not shown. N represents the number of biological replicates.

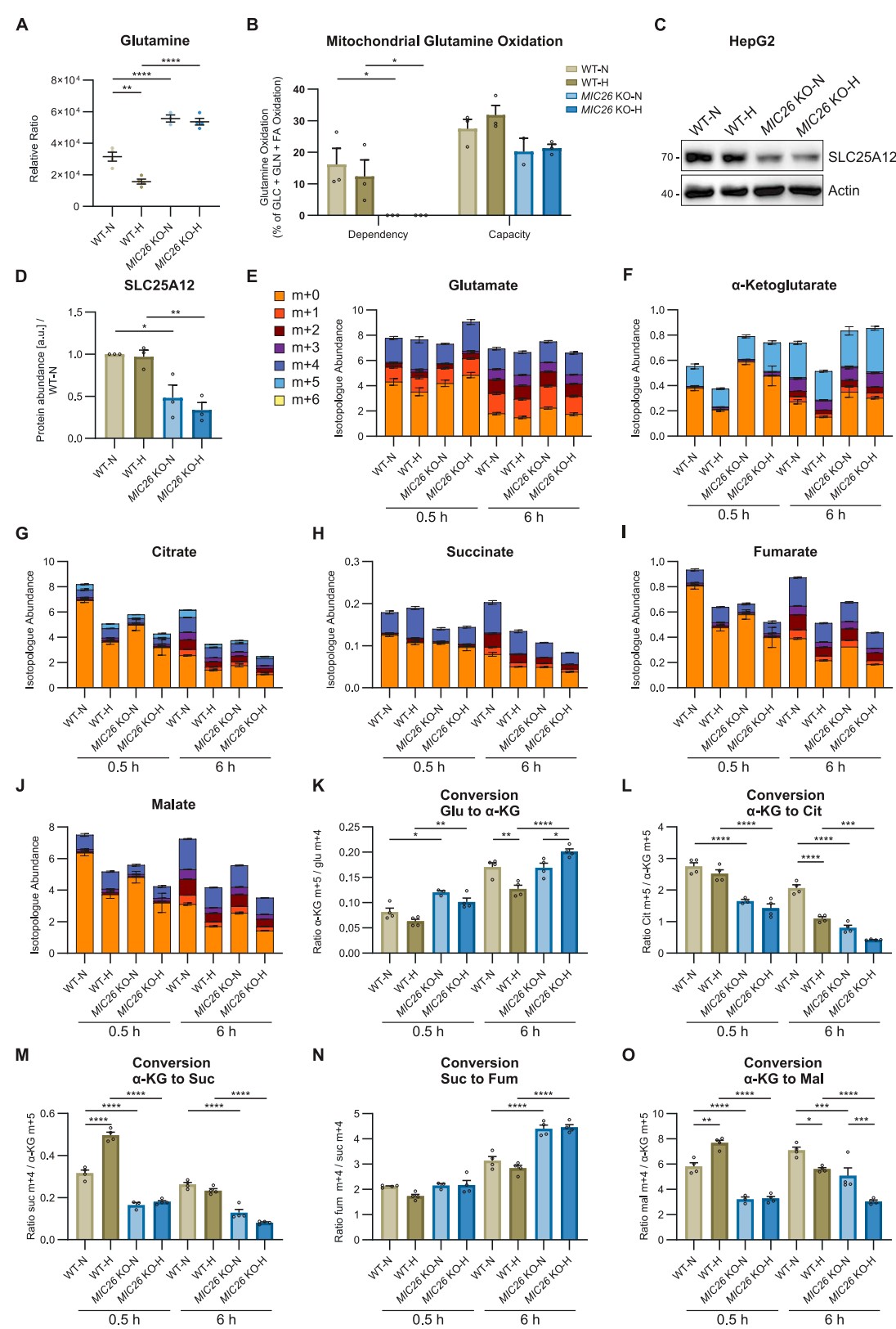

**Figure 6. Aberrant glutamine metabolism is observed in *MIC26* KOs independent of nutritional status.**
**(A)** Metabolomics analysis (GC-MS) shows that glutamine levels were strongly increased in *MIC26* KO cells cultured in both normo- and hyperglycemia at steady state compared with respective WT (N = 3–4). **(B)** Quantification of mitochondrial glutamine dependency and capacity analysis, using Seahorse XF analyzer mito fuel flex test assay, shows a diminished mitochondrial respiratory dependency on glutamine. A non-significant mitochondrial respiratory decreased capacity of *MIC26* KO cells was

*MIC26* KO cells by using a mitochondrial oxygen consumption assay (Fig 7E). We observed an increased basal respiration in *MIC26* KOs in both normoglycemia and hyperglycemia compared with the respective WT (Fig 7F) unlike *MIC27* and *MIC19* KO cells (Fig S11E and F). The ATP production was increased in only *MIC26* KO cells in hyperglycemia compared with WT-H (Fig S11C) and not in *MIC27* and *MIC19* KO cells (Fig S11H). In addition, decreased SRC was observed in *MIC26* KO-N when compared with WT-N condition (Fig S11D) but not in *MIC27* and *MIC19* KO cells (Fig S11G). Overall, *MIC26* KOs specifically, and not *MIC27* KO and *MIC19* KO cells, demonstrate higher basal respiration in both nutrient conditions. To elucidate the increased basal respiration, we performed blue native PAGE to understand the assembly of OXPHOS complexes along with in-gel activity assays (Fig 7G). *MIC26* deletion consistently led to an increase in the levels of OXPHOS complexes I, III, IV and dimeric and oligomeric complex V (shown in green arrows) (Fig 7G, left blots, respectively, for each complex). The increased assembly of OXPHOS complexes was also accompanied by respective increase in in-gel activity (shown in blue arrows) (Fig 7G, right blots, respectively, for each complex). This is consistent with the previously observed increased basal respiration (Fig 7F) and the succinate to fumarate conversion representing an increased complex II activity (Fig 6N). Altogether, we conclude that formation and stability of OXPHOS (super-) complexes and their activity is dependent on MIC26.

## Discussion

Our study identifies MIC26 as a critical regulator at the crossroads of several major metabolic pathways. Based on detailed multi-omics analyses, we deciphered an intricate interplay between MIC26, a mitochondrial IM protein, and global cellular metabolic adaptations. To understand these metabolic changes and their dependency on mitochondrial ultrastructure and function is of high medical relevance as nutrient overload is known to cause obesity and T2DM in humans. In fact, *MIC26* mutations are also associated with mitochondrial myopathy, lactic acidosis (Beninca et al, 2021), and lethality and progeria-like phenotypes (Peifer-Weiß et al, 2023). We showed that cellular fatty acid synthesis, cholesterol biosynthesis and LD formation is promoted by MIC26 under high glucose conditions but that these pathways are conversely inhibited by MIC26 under normal glucose concentrations (Fig 7H). The important role of MIC26 in channelling nutrient excess from glucose into lipids underscores its reported links to obesity (Tian et al, 2017) and diabetes (Lamant et al, 2006) as it known that ectopic lipid accumulation is a common feature of the development of metabolic diseases including NAFLD and insulin resistance. Moreover,

metabolism of glutamine via glutaminolysis is strongly impaired in the absence of MIC26. First, we discuss, how and why MIC26 promotes lipid anabolism in hyperglycemia and what is known from earlier studies in this context. Previously in mammalian cells, we characterized the role of MIC26, which contained a conserved apolipoprotein A1/A4/E family domain, in regulating mitochondrial ultrastructure and function (Koob et al, 2015). We showed that both an increase and decrease in MIC26 was detrimental to mitochondrial function indicating that optimal MIC26 amounts are essential for cellular homeostasis. Despite the demonstration of an increase in mitochondrial structural proteins, such as MIC60, SAMM50, and MIC19, connected with up-regulation of key metabolic pathways in mice fed with HFD compared with normal diet (Guo et al, 2013), the interplay of MICOS proteins, including MIC26, and metabolism is not clear. Because, classically apolipoproteins bind to lipids and mediate their transport in the bloodstream (Mehta & Shapiro, 2022), the presence of MIC26 in a non-classical environment such as the IM raises various questions about its function. Interestingly, a previous report revealed a connection between increased levels of *MIC26* transcripts and nutrient conditions mimicked by oleic acid treatment (Wu et al, 2013). How does the loss of MIC26 alter central metabolic pathways including lipid metabolism in hyperglycemia? In this study, we found an increase in MIC26 in WT cells cultured in hyperglycemia. Concomitant to MIC26 increase, we found that MIC26 stimulates the formation of LDs when glucose is in excess. We demonstrate that MIC26 is essential for glucose use and channelling glycolytic intermediates towards lipid anabolism regulating the accumulation of the LD content. This is supported by several findings including the determined levels of pyruvate and TCA cycle intermediates indicating that by boosting pyruvate levels, MIC26 further increases the amounts of the TCA cycle metabolites including citrate levels which serve as a precursor for cholesterol and FFA synthesis. This connection between lipid synthesis and MIC26 is further strengthened by earlier reports in the context of diabetes or obese models. Dogs fed with a HFD for 9 wk (Philip-Couderc et al, 2003) and diabetic patients (Lamant et al, 2006) showed increased *Mic26* transcripts in the heart. Increased TAG and DAG were found upon *MIC26* overexpression in murine liver (Tian et al, 2017) and hearts (Turkieh et al, 2014), respectively, showing modulatory roles of MIC26 in lipid metabolism. Our data reveal a major MIC26-dependent alteration of metabolite transporters of the mitochondrial IM and also metabolite levels. Thus, loss of MIC26 either alters the level, the activity, or the submitochondrial distribution of various metabolite transporters. In line with our data, the export of citrate from the mitochondrial matrix to the cytosol is presumably of particular importance. MIC26 could regulate metabolite exchange mechanistically either via protein-

observed compared with respective WT conditions (N = 3). **(C, D)** Western blot analysis (C) along with respective quantification (D) show reduced amounts of the glutamate aspartate antiporter SLC25A12 (ARALAR/AGC1), present in mitochondria, in *MIC26* KO cell lines compared with respective WT cells (N = 3). **(E, F, G, H, I, J)** Representation of labelled (m + 1 – m + 6) and unlabeled (m + 0) species of glutamate (GC-MS) (E), and TCA cycle metabolites (AEC-MS) $\alpha$-KG (F), citrate (G), succinate (H), fumarate (I), and malate (J), from glutamine tracing experiments after labelling for 0.5 and 6 h (N = 4). **(K, L, M, N, O)** Conversion rates from different TCA cycle reactions calculated using the ratio of highest labelled species abundances for the conversions of glutamate to $\alpha$-KG (K), $\alpha$-KG to citrate (L), $\alpha$-KG to succinate (M), succinate to fumarate (N), and $\alpha$-KG to malate (N = 4). **(A, B, D, E, F, G, H, I, J, K, L, M, N, O)** Data are represented as mean ± SEM. Statistical analysis was performed using one-way ANOVA with *$P$ < 0.05, **$P$ < 0.01, ***$P$ < 0.001, ****$P$ < 0.0001 for all meaningful combinations (except comparison of WT-H with *MIC26* KO-N and vice versa). Non-significant $P$-values are not shown. N represents the number of biological replicates.
Source data are available for this figure.

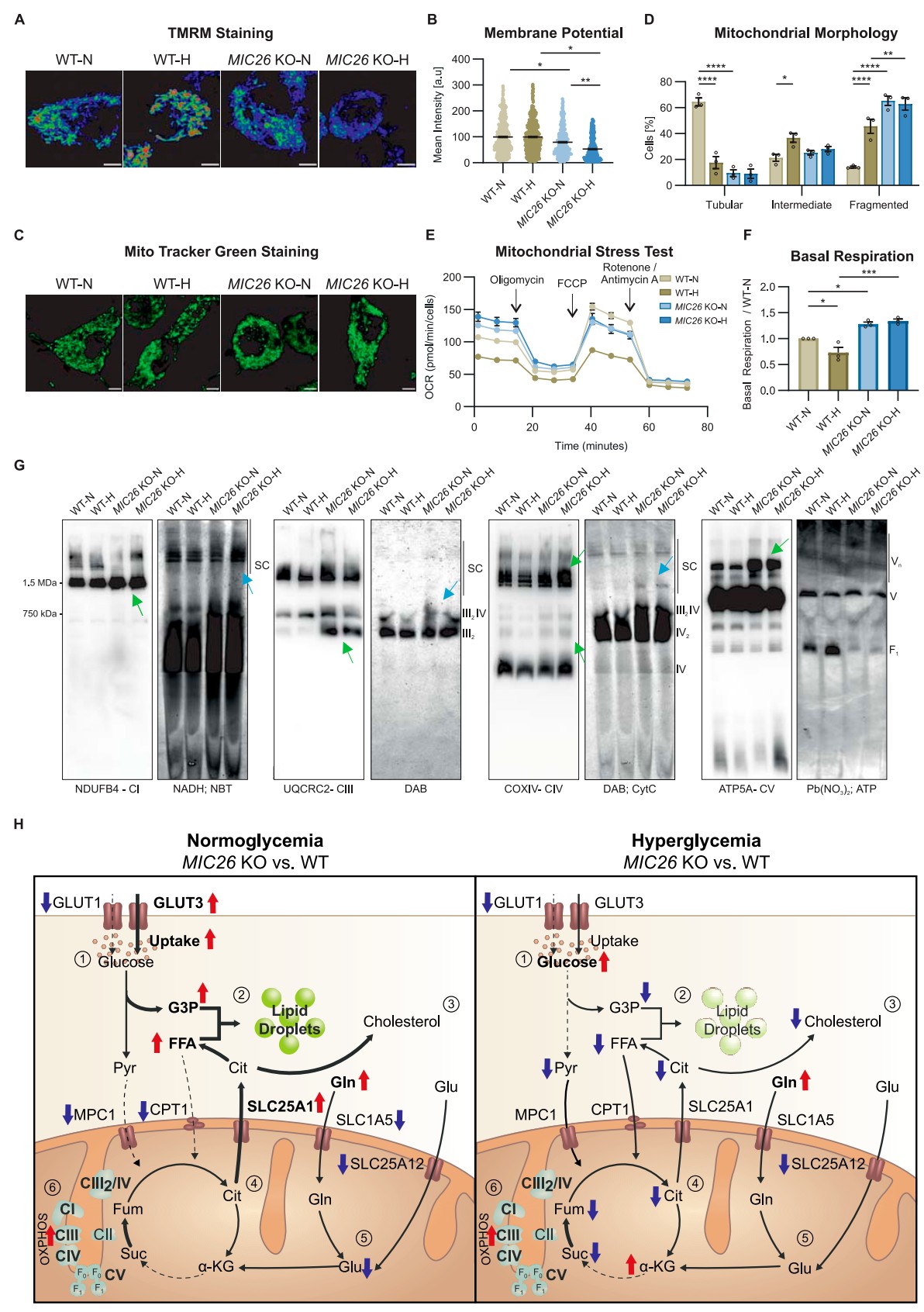

protein interactions of MIC26 to distinct metabolite transporters such as SLC25A12, by altering the accessibility of metabolites to various transporters because of altered cristae morphology. Overall, we propose that MIC26 regulates metabolite exchange between the cytosol and mitochondria and vice versa in a nutrient-dependent manner which is critical for adaptations to excess of glucose.

Under balanced nutrient conditions, MIC26 plays a different role compared with nutrient excess conditions. MIC26 decreases the key enzymes regulating the upper half of glycolytic pathway involved in ATP consumption phase. MIC26 prevents an increase in FFAs and G3P culminating in uncontrolled accumulation of LD content. In line with this, it was recently shown that the loss of MIC26 in BAT led to up-regulation of glycolysis and fatty acid synthesis pathways (Guo et al, 2023). This was accompanied by impaired thermogenic activity of BAT, mitochondrial ultrastructure and function which reiterates the role of MIC26 in metabolic reprogramming. In normoglycemia, we found that the presence of MIC26 leads to a decrease in most of the transcripts of enzymes participating in cholesterol biosynthesis, including the sterol regulatory element binding transcription factor 2 (SREBP2) (Fig S12C) which is a master regulator of genes involved in sterol and fatty acid synthesis (Madison, 2016). However, we observed equal amounts of cholesterol in MIC26 KO and WT cells under normoglycemia. Thus, MIC26 in normoglycemia facilitates equal metabolite distribution to either cholesterol biosynthesis or lipogenesis. This MIC26-mediated metabolic switch based on the amount/type of cellular fuel is essential for maintaining key metabolic pathways. Regarding key metabolic pathways, we used several functional assays to show that the role of MIC26 is not duplicated either by MIC27 or MIC19 indicating a specific role. A balanced amount of MIC26 is essential for how much glucose is channelled into lipid synthesis. In congruence with our observations, key lipid metabolism genes were altered upon Mic26 overexpression with an interesting antagonistic regulation of de novo lipid synthesis genes depending on nutritional conditions (Tian et al, 2017). This study demonstrated an increase in important transcripts regulating lipid synthesis such as ACACA, FASN, and SCD in mice, overexpressing Mic26, fed with normal diet and a decrease when fed with HFD, compared with respective control mice. We further observed a decrease in CPT1A level and activity in MIC26 KO cell lines and WT-H. CPT1A activity is known to be regulated either on a transcriptional level via

peroxisome proliferator-activated receptor α (PPARα) and peroxisome proliferator-activated receptor gamma coactivator 1 alpha (PGC-1α) or by allosteric inhibition through malonyl-CoA (López-Viñas et al, 2007; Song et al, 2010). A recent study demonstrated the down-regulation of PPARα protein level in BAT in adipose tissue-specific MIC26 KO mice (Guo et al, 2023). Furthermore, we observed a high up-regulation of ACACA enzyme, which converts acetyl-CoA to malonyl-CoA during de novo lipogenesis. Accordingly, it is possible that MIC26 deletion in normoglycemia could on one hand reduce the expression of PPARα leading to decreased CPT1A expression and on the other hand increase malonyl-CoA formation leading to decreased CPT1A activity. Taken together, we used a multi-omics approach and a variety of functional assays to decipher that loss of MIC26 leads to an antagonistic regulation of glycolysis, lipid and cholesterol synthesis dependent on cellular nutritional stimulation.

Besides the antagonistic regulation mediated by MIC26 in different nutrient conditions, there are general roles of MIC26 in metabolic pathways which are independent of nutrient conditions. Among the MICOS proteins, proteins such as MIC60 are considered as core components as MIC60 deletion leads to a consistent loss of CJs (Kondadi et al, 2020a; Stephan et al, 2020), whereas the effect of MIC26 deletion on loss of CJs varies with the cell line. Loss of CJs was observed in 143B (Koob et al, 2015) and HAP1 cells (Anand et al, 2020) in contrast to HeLa cells (Stephan et al, 2020). MIC26 deletion in HepG2 cells in this study revealed a significant reduction in CJs when normalized to the cristae number, highlighting a possible major role of MIC26 in liver-derived cell lines. Concomitant to the reduction in CJs, we also observed alterations of vital transporters in the mitochondrial IM and OM. It was recently described that deletion of stomatin-like protein 2 (SLP2) leads to a drastic MIC26 degradation mediated by the YME1L protease (Naha et al, 2023 Preprint). SLP2 was proposed as a membrane scaffold for PARL and YME1L named as the SPY complex (Wai et al, 2016). It is therefore conceivable that MIC26 could be concentrated in lipid-enriched nanodomains of the IM justifying its apolipoprotein nomenclature. When we checked the mitochondrial function upon MIC26 deletion in HepG2 cells, we found that RCCs have enhanced respiratory capacity which was because of: (a) increase in the levels of native RCCs and supercomplexes and (b) increase in the activity of RCCs corresponding to increased RCC amounts. Thus, MIC26 could perform structural and functional roles which may or may not be mutually exclusive to the MICOS complex. A

**Figure 7. MIC26 regulates mitochondrial bioenergetics by restricting the ETC activity and OXPHOS (super-)complex formation.**
**(A, B)** Representative pseudocolour rainbow LUT intensities from confocal images of WT and MIC26 KO HepG2 cells stained with TMRM show a reduction in ΔΨ$_m$ upon MIC26 deletion in both normoglycemia and hyperglycemia when compared with respective WT cells (A). **(B)** Quantification represents mean TMRM fluorescence intensity per cell normalized to mean intensity of WT-N in all biological replicates (B) (N = 3). Scale bar represents 5 μm in (A, C). **(C, D)** Representative confocal images of mitochondrial morphology, visualized by MitoTracker green staining (C), show that loss of MIC26 shifts mitochondrial morphology from tubular mitochondrial network in WT normoglycemic conditions to fragmented phenotype irrespective of supplemented glucose amount (D) (N = 3). Scale bar represents 5 μm. **(E, F)** Representative mitochondrial stress test with Seahorse XF analyzer, with sequential injection of oligomycin, FCCP and rotenone/antimycin (E) (n = 19–23). **(F)** Quantification from various biological replicates shows a significant increase in basal respiration in MIC26 KOs cultured in both normo- and hyperglycemia (F) (N = 3). **(G)** Blue native (respective left panel) and clear native (respective right panel) PAGE analysis reveals an overall increase in OXPHOS complex formation (for CI, CIII, CIV, and CV, green arrows) and corresponding increases in-gel activity of supercomplexes, and complex III$_2$IV (blue arrows) upon MIC26 deletion. CV shows no in-gel activity alterations whereas a decreased in-gel activity of F$_1$ occurs upon loss of MIC26. Native PAGEs were performed in three biological replicates and representative gels are shown. **(H)** Model representing the antagonistic regulation of metabolic pathways encompassing glucose usage, lipid droplet formation, cholesterol synthesis, and decrease in TCA cycle metabolites in MIC26 deficient HepG2 cells dependent on nutritional conditions compared with respective WT cells. An increase in glutamine levels and assembly of various OXPHOS complexes is observed in MIC26 KOs independent of the nutritional status. Arrows indicate respective up (red) or down-regulated (blue) protein/metabolite or activity levels, respectively. In the model, left panel indicates normoglycemic, whereas the right panel represents the hyperglycemic conditions. **(B, D, E, F)** Data are represented as mean ± SEM. Statistical analysis was performed using one-way ANOVA with *$P < 0.05$, **$P < 0.01$, ***$P < 0.001$, ****$P < 0.0001$ for all meaningful combinations (except comparison of WT-H with MIC26 KO-N and vice versa). Non-significant P-values are not shown. N represents the number of biological replicates and n the number of technical replicates.

reduction in SLC25A12, an antiporter of cytoplasmic glutamate and mitochondrial aspartate, which is present in the IM was observed upon *MIC26* deletion independent of the nutritional status. We also found that SLC25A12 could be an interactor of MIC26 upon using standard interaction databases available online. Presumably, the interaction of SLC25A12 with MIC26 is important for the stability of the former. Such an intricate relationship between MIC26 and metabolite transporters in the IM makes it tempting to speculate that mitochondrial membrane remodelling is linked to its metabolic function. In fact, a closer look at the mitochondrial carrier family (SLC25) transcriptomics and proteomics data sets revealed that most of the SLC25 transporters were differentially regulated upon *MIC26* deletion in normoglycemia and hyperglycemia (Fig S12A and B). A prominent example is the reduction in SLC1A5 in *MIC26* KO when compared with WT in hyperglycemia and normoglycemia. A recent study showed that a variant of SLC1A5 present in the mitochondrial IM is responsible for transporting glutamine into the mitochondria (Yoo et al, 2020a). *MIC26* deletion leading to reduced total amounts of SLC1A5 also indicates reduced transport of glutamine into mitochondria. This was in line with an accumulation of glutamine upon loss of MIC26 at steady state. However, using glutamine tracing experiments we did not observe a decrease in labelled glutamate species but observed an accumulation of α-KG. This could be because of the observed decrease in α-KG dehydrogenase resulting in reduced conversion of α-KG to other metabolites of the TCA cycle in particular under high glucose conditions. On the other hand, besides mitochondrial glutamine usage to fuel the TCA cycle, glutamine is known to be an essential source for nucleotide biosynthesis (Yoo et al, 2020b). *MIC26* KO cells showed a decreased growth rate (Fig S12D–F). Hence, glutamine accumulation and with that reduced conversion to nucleotides is a possible mechanism leading to growth deficiencies of *MIC26* KO cells. We found that the transcripts and protein levels of NUBPL were prominently down-regulated upon *MIC26* deletion independent of the glucose concentrations of the cell culture media (Fig S12G and H). NUBPL was demonstrated to function as an assembly factor for complex I (Sheftel et al, 2009). Despite the prominent reduction in NUBPL, we did not find any discrepancy in complex I assembly or its activity most likely because of increased RCC amounts. We also found that the transcripts and protein levels of DHRS2 were significantly reduced in *MIC26* KO (Fig S12I and J). DHRS2

is implicated in reprogramming of lipid metabolism (Li et al, 2021) and was found to be down-regulated in T2DM (De Silva et al, 2022).

We further found that hyperglycemia and *MIC26* deletion resulted in a fragmented mitochondrial morphology compared with WT-N. Mitochondrial dynamics and cellular metabolism including nutritional demands are closely interlinked (Mishra & Chan, 2016). Nutritional overload was associated with increased mitochondrial fragmentation (Yu et al, 2006), whereas starvation led to formation of a tubular mitochondrial network (Gomes et al, 2011). Furthermore, mice lacking the ability to undergo mitochondrial fission by liver specific deletion of *Drp1* were protected from lipid accumulation in the liver and insulin resistance upon HFD feeding (Wang et al, 2015). Obesity is associated with increased mitochondrial fragmentation in multiple studies. A recent study showed that mitochondrial fragmentation is positively correlated to mitochondrial long-chain FFA oxidation capacity via an increased activity of CPT1A (Ngo et al, 2023). A stronger membrane curvature resulting from mitochondrial fragmentation induces a conformational change leading to a decreased inhibitory binding ability of malonyl-CoA on CPT1 activity. Even though we observed mitochondrial fragmentation upon *MIC26* deletion, we did not observe increased FAO. This discrepancy could be explained by a reduction in CPT1A amount on one hand and likely increased production of malonyl-CoA on the other hand because of increased amounts of SLC25A1 and ACACA participating in fatty acid synthesis. Hence, deletion of MIC26, leading to mitochondrial fragmentation, contributes to ectopic cellular lipid accumulation but not FAO.

In sum, under balanced nutrient availability, we provide evidence that MIC26 is important to allow efficient metabolite channelling, mainly via glycolysis, thereby preventing unwanted channelling into lipogenesis. In addition, MIC26 is important to promote exactly the latter when glucose is in excess. This is important for cells to adapt to nutrient overload and explains earlier reports linking MIC26 to diabetes. We propose that MIC26 acts as a sensor and valve that opens towards lipid synthesis only when glucose is in excess. Future studies will have to decipher how changes in IM structure directly affect metabolite exchange and how this is regulated dynamically.

# Materials and Methods

**Key resources table.**

| Reagent or resource | Source | Identifier |
|---|---|---|
| Antibodies | | |
| MIC26 | Invitrogen | Cat# PA5-116197 RRID: AB_2900831 |
| MIC27 | Sigma-Aldrich | Cat# HPA000612 RRID: AB_1078594 |
| MIC10 | Abcam | Cat# ab84969 RRID: AB_1924831 |
| MIC13 | Pineda | Custom-made |
| MIC25 | Protein tech | Cat# 20639-1-AP RRID: AB_1069767 |
| MIC60 | Abcam | Cat# ab110329 RRID: AB_10859613 |
| MIC19 | Protein tech | Cat# 25625-1-AP RRID: AB_2687533 |
| MFN1 | Santa Cruz Biotechnologies | Cat# sc-50330 RRID: AB_2250540 |

**Continued**

| Reagent or resource | Source | Identifier |
|---|---|---|
| MFN2 | Abcam | Cat# ab50838 RRID: AB_881507 |
| OPA1 | Pineda | Custom-made |
| DRP1 | Cell Signaling Technologies | Cat# 5391 RRID: AB_11178938 |
| β-Actin | Invitrogen | Cat# MA1-744 RRID: AB_2223496 |
| HSP60 | Sigma-Aldrich | Cat# SAB4501464 RRID: AB_10746162 |
| CPT1A | Proteintech | Cat# 15184-1-AP RRID: AB_2084676 |
| Goat IgG anti-mouse IgG | Abcam | Cat# ab97023 RRID: AB_10679675 |
| Goat IgG anti-rabbit IgG | Dianova | Cat# SBA-4050-05 RRID: AB_2795955 |
| NDUFB4 | Abcam | Cat# ab110243 RRID: AB_10890994 |
| UQCRC2 | Abcam | Cat# ab14745 RRID: AB_2213640 |
| COXIV | Abcam | Cat# ab16056 RRID: AB_443304 |
| ATP5A | Abcam | Cat# ab14748 RRID: AB_301447 |
| SLC25A12 | Santa Cruz Biotechnologies | Cat# sc-271056 RRID: AB10608837 |
| **Bacterial and virus strains** | | |
| N/A | | |
| **Biological samples** | | |
| N/A | | |
| **Chemicals, peptides, and recombinant proteins** | | |
| Etomoxir | Sigma-Aldrich | Cat# E1905; CAS: 828934-41-4 |
| 3,3′-diaminobenzidine tetrahydrochloride | Sigma-Aldrich | Cat# 32750; CAS: 868272-85-9 |
| Horse heart cytochrome c | Thermo Fisher Scientific | Cat# 147530010; CAS: 9007-43-6 |
| NADH | Biomol | Cat# 16132.1 CAS: 606-68-8 |
| Nitroblue tetrazolium chloride | Biomol | Cat# 06428.1 CAS: 298-83-9 |
| ATP | Sigma-Aldrich | Cat# A7699 CAS: 34369-07-8 |
| Lead (II) nitrate/$Pb(NO_3)_2$ | Sigma-Aldrich | Cat# 1073980100 CAS: 10099-74-8 |
| MitoTracker Green | Invitrogen | Cat# M7514 CAS: 201860-17-5 |
| TMRM | Invitrogen | Cat# T668 |
| BODIPY 493/503 | Cayman Chemicals | Cat# Cay25892-5 CAS: 121207-31-6 |
| Poly-D-lysine | Sigma-Aldrich | Cat# P7886 CAS: 2796-99-4 |
| Stable glutamine | PAN-Biotech | Cat# P04-82100 |
| PenStrep | PAN-Biotech | Cat# P06-07100 |
| BSA | Biomol | Cat#Cay29556 |
| Palmitate-BSA | Biomol | Cat#Cay29558 |
| L-carnitine | Sigma-Aldrich | Cat#C0283 CAS: 645-46-1 |
| Ribitol | Sigma-Aldrich | Cat# A5502 |
| L-glutamine-13C5-15N2 | Sigma-Aldrich | Cat# 607983 CAS: 607983 |
| **Critical commercial assays** | | |
| Glycolysis Stress Test | Agilent Technologies | Cat# 103020-100 |
| Mito Stress Test | Agilent Technologies | Cat# 103015-100 |
| Mito Fuel Flex Test | Agilent Technologies | Cat# 103260-100 |
| Glucose Uptake-Glo | Promega | Cat# J1241 |
| GoScript Reverse Transcription Mix, Oligo(dT) | Promega | Cat# A2791 |

| Reagent or resource | Source | Identifier |
|---|---|---|
| GoTaq qPCR Master Mix | Promega | Cat# A6002 |
| RNeasy Mini Kit | QIAGEN | Cat# 74106 |
| **Deposited data** | | |
| MIC26 interactome | N/A | Nextprot, Intact BioGrid |
| **Experimental models: cell lines** | | |
| HepG2 | Sigma-Aldrich | Cat# 85011430 RRID: CVCL_0027 |
| **Experimental models: organisms/strains** | | |
| N/A | | |
| **Oligonucleotides** | | |
| Primer CPT1A Forward GATCCTGGACAATACCTCGGAG | This paper | N/A |
| Primer CPT1A Reverse CTCCACAGCATCAAGAGACTGC | This paper | N/A |
| Primer HPRT1 Forward 5'-CATTATGCTGAGGATTTGGAAAGG-3' | This paper | N/A |
| Primer HPRT1 Reverse 5'-CTTGAGCACACAGAGGGCTACA-3' | This paper | N/A |
| **Recombinant DNA** | | |
| Double Nickase Plasmid MIC26 KO | Santa Cruz Biotechnology | Cat# sc-413137-NIC |
| **Software and algorithms** | | |
| Prism | GraphPad | RRID: SCR_002798 |
| Seahorse Wave | Agilent Technologies | RRID: SCR_014526 |
| R Studio | Posit PBC | RRID: SCR_000432 |
| Cytoscape | Cytoscape Consortium | RRID: SCR_003032 |
| Volocity 3D Image Analysis Software | Perkin Elmer | RRID: SCR_002668 |
| MassHunter Qualitative | Agilent Technologies | RRID: SCR_015040 |
| BCL Convert Tool | Illumina | N/A |
| CLC Genomics Workbench | QIAGEN | RRID: SCR_017396 |
| CLC Gene Set Enrichment Test | QIAGEN | RRID: SCR_003199 |
| Proteome Discoverer | Thermo Fisher Scientific | RRID: SCR_014477 |
| **Other** | | |
| DMEM 1 g/liter glucose | PAN-Biotech | Cat# P04-01500 |
| DMEM 4.5 g/liter glucose | PAN-Biotech | Cat# P04-82100 |
| FBS | Capricorn Scientific | Cat# FBS-11A |

## Cell culture and treatment conditions

HepG2 cells were cultured in 1 g/liter glucose DMEM (PAN-Biotech) supplemented with 10% FBS (Capricorn Scientific), 2 mM stable glutamine (PAN-Biotech) and penstrep (PAN-Biotech, penicillin 100 and 100 µg/ml streptomycin). Cells were grown at 37°C supplied with 5% CO$_2$. HepG2 *MIC26* KO, *MIC27* KO, and *MIC19* KO cells were generated using the double nickase method as described before (Lubeck et al, 2023). The plasmid mix was commercially available (Santa Cruz Biotechnology): MIC26: double nickase plasmid sc-413137-NIC, MIC27 double nickase plasmid sc-414464-NIC, and MIC19 double nickase plasmid sc-408682-NIC. Cells cultured in standard growth media were divided equally into two cell culture flasks and grown in either 1 g/liter glucose DMEM (normoglycemia)

or 4.5 g/liter glucose DMEM (hyperglycemia) (PAN-Biotech) supplemented with above-mentioned reagents. Cells were cultured in normoglycemia and hyperglycemia for a prolonged duration of 3 wk. During the 3 wk, cell splitting was carried out twice a week with the corresponding media.

## SDS gel electrophoresis and Western blotting

After three washes with 2 ml DPBS (PAN-Biotech), the cells were harvested by scraping and resuspending in an appropriate volume of RIPA buffer (150 mM NaCl, 0.1% SDS, 0.05% Sodium deoxycholate, 1% Triton-X-100, 1 mM EDTA, 1 mM Tris, pH 7.4, 1x protease inhibitor (Sigma-Aldrich), PhosSTOP (Roche). Protein concentration was determined using DC protein assay Kit (5000116; Bio-Rad). SDS

samples were prepared with Laemmli buffer and heated for 5 min at 95°C. Depending on the proteins investigated, a variety of SDS electrophoresis gels (8%, 10%, 12% or 15%) were used for running and separating protein samples. Subsequently, proteins were transferred onto nitrocellulose membranes and stained using Ponceau S (Sigma-Aldrich). After destaining, nitrocellulose membranes were blocked with 5% milk in 1x TBS-T for 1 h, washed three times with TBS-T and probed at 4°C overnight with the following primary antibodies: MIC26 (1:1,000; Invitrogen), MIC27 (1:2,000; Sigma-Aldrich), MIC10 (1:1,000; Abcam), MIC13 (1:1,000; Pineda custom-made), MIC25 (1:1,000; Proteintech), MIC60 (1:1,000; Abcam), MIC19 (1:1,000; Proteintech), MFN1 (1:1,000; Santa Cruz Biotechnologies), MFN2 (1:1,000; Abcam), OPA1 (1:1,000; Pineda custom-made), DRP1 (1:1,000; Cell Signaling Technology), $\beta$-Actin (1:2,000; Invitrogen), HSP60 (1:2,000; Sigma-Aldrich), and CPT1A (1:1,000; Proteintech). Goat IgG anti-Mouse IgG (1:10,000; Abcam) and Goat IgG anti-Rabbit IgG (1:10,000; Dianova) conjugated to HRP were used as secondary antibodies. The chemiluminescent signals were obtained using Signal Fire ECL reagent (Cell Signaling Technology) and VILBER LOURMAT Fusion SL equipment (Peqlab).

## Blue native and clear native PAGE

$5 \times 10^6$ HepG2 cells were seeded onto 15 cm dishes and cell culture medium was replaced every 2 d until 80% confluency was reached. Cells were washed three times with cold PBS, scraped and pelleted at 900$g$, 4°C for 5 min. Cell pellets were resuspended in 1 ml lysis buffer (210 mM mannitol, 70 mM sucrose, 1 mM EDTA, 20 mM Hepes, 0,1% BSA, 1x protease inhibitor) and incubated on ice for 10 min. Mitochondria were isolated by repetitive strokes of mechanical disruption using a 20G canula and sequential centrifugation steps at 1,000$g$, 4°C for 10 min to remove cell debris and 10,000$g$, 4°C for 15 min to pellet mitochondria. Mitochondrial pellet was resuspended in BSA-free lysis buffer and protein concentration was determined using DC Protein Assay Kit.

For blue native page, 100 $\mu$g of mitochondria was solubilized for 1 h on ice using 2.5 g/g of digitonin to protein ratio. The samples were centrifuged for 20 min at 20,000$g$ and 4°C to pellet insolubilized material. The supernatants were supplemented with loading buffer (50% glycerol, 8 g/g Coomassie to detergent ratio) and immediately loaded onto 3–13% gradient gel. Complexes were separated at 150 V, 15 mA for 16 h. Thereafter, protein complexes were transferred onto PVDF membrane and blocked overnight with 5% milk in TBS-T at 4°C. For identification of relevant protein complexes, the membranes were decorated with the following antibodies: NDUFB4 (Abcam, 1:1,000), UQCRC2 (1:1,000; Abcam), COXIV (1:1,000; Abcam), ATP5A (1:1,000; Abcam), goat IgG anti-mouse IgG (1:10,000; Abcam), and goat IgG anti-rabbit IgG (1:10,000; Dianova) conjugated to HRP. The chemiluminescent signals were obtained using Pierce SuperSignal West Pico PLUS chemiluminescent substrate reagent (Thermo Fisher Scientific) and VILBER LOURMAT Fusion SL equipment (Peqlab).

For clear native gels, 300 $\mu$g mitochondria were solubilized on ice for 1 h with 2.5 g/g digitonin to protein ratio. The samples were centrifuged for 20 min at 20,000$g$ and 4°C to pellet insolubilized material. The supernatants were supplemented with loading buffer (50% glycerol, 0.01% Ponceau S) and immediately loaded onto 3–13% gradient gels. Complexes were separated at 150 V, 15 mA for

16 h. To assess complex in-gel activity, the gel slices were incubated in respective buffer solutions for several hours at RT. For complex I activity, the gel was incubated in 5 mM Tris–HCl (pH 7.4), 0.1 mg/ml NADH and 2.5 mg/ml nitro blue tetrazolium chloride (NBT). For complex III, the gel was incubated in 50 mM sodium phosphate buffer (pH 7.2), 0.1% 3,3′-diaminobenzidine tetrahydrochloride (DAB). To assess complex IV activity, the gel was incubated in 50 mM sodium phosphate buffer (pH 7.2), 0.05% DAB and 50 $\mu$M horse heart cytochrome $c$, and for complex V, the gel was incubated in 35 mM Tris-base, 270 mM glycine, 14 mM MgSO$_4$, 0.2% (wt/vol) Pb(NO$_3$)$_2$, and 8 mM ATP.

## RNA isolation and quantification

Total RNA was extracted from cell pellets using RNeasy Mini Kit (QIAGEN) according to the manufacturer's protocol. RNA quality and quantity were assessed using BioSpectrometer (Eppendorf). cDNA synthesis from 5 $\mu$g RNA was performed using the GoScript Reverse Transcriptase Kit (Promega). Next, quantitative real-time PCR was performed in Rotor Gene 6000 (Corbett Research) using GoTagR qPCR Master Mix (Promega) according to manufacturer's instructions with the following primers:

### *CPT1A*
Forward: 5′-GATCCTGGACAATACCTCGGAGC-3′
 Reverse: 5′-CTCCACAGCATCAAGAGACTGC-3′

### HPRT1 *(housekeeping gene)*
Forward: 5′-CATTATGCTGAGGATTTGGAAAGG-3′
 Reverse: 5′-CTTGAGCACACAGAGGGCTACA-3′
 $C_t$ values were normalized to housekeeping gene *HPRT1* followed by normalization of $\Delta C_t$ values to average $\Delta C_t$ of WT-N control group.

## Transcriptomics

Cells were seeded in quadruplicates onto 10 cm dishes in corresponding cell culture media and medium was replaced every 2 d until 80% cell confluency was obtained. For preparation of RNA, cells were washed three times with cold PBS and subsequently scraped and pelleted. RNA isolation from cell pellets was performed using RNeasy Mini Kit (QIAGEN) including DNase digestion according to the manufacturer's protocol. Sample concentration was determined and 1 $\mu$g RNA was aliquoted for transcriptomics analysis. Total RNA samples were quantified (Qubit RNA HS Assay; Thermo Fisher Scientific) and quality measured by capillary electrophoresis using the Fragment Analyzer and the "Total RNA Standard Sensitivity Assay" (Agilent Technologies, Inc.). All samples in this study showed RNA Quality Numbers (RQN) with a mean of 10.0. The library preparation was performed according to the manufacturer's protocol using the "VAHTS Stranded mRNA-Seq Library Prep Kit" for Illumina. Briefly, 700 ng total RNA were used as input for mRNA capturing, fragmentation, the synthesis of cDNA, adapter ligation and library amplification. Bead purified libraries were normalized and finally sequenced on the NextSeq2000 system (Illumina Inc.) with a read setup of 1 × 100 bp. The BCL Convert Tool (version 3.8.4) was used to convert the bcl files to fastq files as well for adapter trimming and demultiplexing.

Data analyses on fastq files were conducted with CLC Genomics Workbench (version 22.0.2; QIAGEN). The reads of all probes were adapter trimmed (Illumina TruSeq) and quality trimmed (using the default parameters: bases below Q13 were trimmed from the end of the reads, ambiguous nucleotides maximal 2). Mapping was performed against the *Homo sapiens* (hg38; GRCh38.107) (20 July 2022) genome sequence. After grouping of samples (four biological replicates) according to their respective experimental conditions, the statistical differential expression was determined using the CLC differential expression for RNA-Seq tool (version 2.6; QIAGEN). The resulting *P*-values were corrected for multiple testing by FDR and Bonferroni correction. A *P*-value of ≤0.05 was considered significant. The CLC gene set enrichment test (version 1.2; QIAGEN) was performed with default parameters and based on the GO term "biological process" (*H. sapiens*; 01 May 2021).

The data discussed in this publication have been deposited in NCBI's Gene Expression Omnibus (Edgar et al, 2002) and are accessible through GEO Series accession number GSE248848.

## Proteomics

Cells were seeded in quintuplicates onto 10 cm dishes in corresponding cell culture media and medium was replaced every 2 d until 80% cell confluency was obtained. Cells were washed four times with PBS, scraped and pelleted in a pre-weighed Eppendorf tube. After complete removal of PBS, cells were immediately frozen in liquid nitrogen and sample weight was determined for normalization. Proteins were extracted from frozen cell pellets as described elsewhere (Poschmann et al, 2014). Briefly, cells were lysed and homogenized in urea buffer with a TissueLyser (QIAGEN) and supernatants were collected after centrifugation for 15 min at 14,000$g$ and 4°C. Protein concentration was determined by means of Pierce 660 nm protein assay (Thermo Fisher Scientific). For LC-MS analysis, a modified magnetic bead-based sample preparation protocol according to Hughes and colleagues was applied (Hughes et al, 2019). Briefly, 20 $\mu$g total protein per sample was reduced by adding 10 $\mu$l 100 mM DTT (dithiothreitol) and shaking for 20 min at 56°C and 1,000 rpm (ThermoMixer C from Eppendorf), followed by alkylation with the addition of 13 $\mu$l 300 mM IAA and incubation for 15 min in the dark. A 20 $\mu$g/$\mu$l bead stock of 1:1 Sera-Mag Speed-Beads was freshly prepared and 10 $\mu$l was added to each sample. Afterwards, 84 $\mu$l ethanol was added and incubated for 15 min at 24°C. After three rinsing steps with 80% EtOH and one rinsing step with 100% ACN, beads were resuspended in 50 mM TEAB buffer and digested with final 1:50 trypsin at 37°C and 1,000 rpm overnight. Extra-digestion was carried out by adding trypsin (final 1:50) and shaking at 37°C and 1,000 rpm for 4 h. The supernatants were collected and 500 ng of each sample digest was subjected to LC-MS.

For the LC-MS acquisition, an Orbitrap Fusion Lumos Tribrid Mass Spectrometer (Thermo Fisher Scientific) coupled to an Ultimate 3000 Rapid Separation liquid chromatography system (Thermo Fisher Scientific) equipped with an Acclaim PepMap 100 C18 column (75 $\mu$m inner diameter, 25 cm length, 2 $\mu$m particle size from Thermo Fisher Scientific) as separation column and an Acclaim PepMap 100 C18 column (75 $\mu$m inner diameter, 2 cm length, 3 $\mu$m particle size from Thermo Fisher Scientific) as trap column was used. A LC-gradient of 180 min was applied. Survey scans were carried out over

a mass range from 200-2,000 m/z at a resolution of 120,000. The target value for the automatic gain control was 250,000 and the maximum fill time 60 ms. Within a cycle time of 2 s, the most intense peptide ions (excluding singly charged ions) were selected for fragmentation. Peptide fragments were analyzed in the ion trap using a maximal fill time of 50 ms and automatic gain control target value of 10,000 operating in rapid mode. Already fragmented ions were excluded for fragmentation for 60 s.

Data analysis was performed with Proteome Discoverer (version 2.4.1.15, Thermo Fisher Scientific). All RAW files were searched against the human Swissprot database (Download: 23.01.2020) and the Maxquant Contaminant database (Download: 20.02.2021), applying a precursor mass tolerance of 10 ppm and a mass tolerance of 0.6 Da for fragment spectra. Methionine oxidation, N-terminal acetylation, N-terminal methionine loss and N-terminal methionine loss combined with acetylation were considered as variable modifications, carbamidomethylation as static modification and tryptic cleavage specificity with a maximum of two missed cleavage sites. Label-free quantification was performed using standard parameters within the predefined workflow. Post processing, proteins were filtered to 1% FDR and a minimum of 2 identified peptides per protein. The mass spectrometry proteomics data have been deposited to the ProteomeXchange Consortium via the PRIDE (Perez-Riverol et al, 2022) partner repository with the dataset identifier PXD047246.

## Metabolomics

Metabolites were analyzed by gas chromatography (GC) and anion exchange chromatography coupled to mass spectrometry (MS). 1.5 × 10$^6$ cells were seeded in quadruplicates onto 6 cm dishes and cultured in the corresponding media overnight. For glutamine tracing experiments, medium was replaced with corresponding growth media containing 2 mM labelled glutamine [U-$^{13}$C$_5$, $^{15}$N$_2$] (Sigma-Aldrich) either for 30 min or 6 h before cell harvesting. For metabolite extraction, cells were washed five times with ice-cold isotonic NaCl solution (0.9%), followed by scraping of cells in 1 ml ice-cold MeOH. Cells were transferred to a 15 ml tube and diluted with 1 ml MilliQ water. Cell suspension was immediately frozen in liquid nitrogen. After thawing on ice, 0.5 ml MilliQ water was added supplemented with 10 $\mu$M internal standard ribitol (Sigma-Aldrich) for polar metabolite analysis. After that 1.5 ml MTBE was added containing 5.4 $\mu$l heptadecanoic acid (1 mg/ml) as internal standard for free fatty acid analysis. After repetitive mixing, samples were incubated on ice for 10 min. Subsequently, polar and nonpolar phases were separated by centrifugation at 4,000$g$ for 10 min at 4°C. The apolar phase was collected, frozen at −80°C and used for free fatty acid analysis. The aqueous phase was diluted with MilliQ water to decrease the organic proportion below 15%. The sample was then frozen at −80°C, dried by lyophilization reconstituted in 500 $\mu$l MilliQ water and filtered before analysis.

For GC-MS, 100 $\mu$l was dried by vacuum filtration. Metabolite analysis was conducted using a 7890B gas chromatography system connected to a 7200 QTOF mass spectrometer (Agilent Technologies) as described previously (Shim et al, 2019). In brief, methoxyamine hydrochloride and N-methyl-N-(trimethylsilyl)trifluoroacetamide were subsequently added to the dried sample to derivatize

functional groups of polar compounds. With an injection volume of 1 $\mu l$, samples were introduced into the GC-MS system and compounds were separated on a HP-5MS column (30 m length, 0.25 mm internal diameter, and 0.25 $\mu m$ film thickness). The software Mass-Hunter Qualitative (v b08; Agilent Technologies) was used for compound identification by comparing mass spectra to an in-house library of authentic standards and to the NIST14 Mass Spectral Library (https://www.nist.gov/srd/nist-standard-reference-database-1a-v14). Peak areas were integrated using MassHunter Quantitative (v b08; Agilent Technologies) and normalized to the internal standard ribitol and cell number. To determine the $^{13}C$ and $^{15}N$ incorporation, isotopologues for individual fragments were analyzed according to the number of possible incorporation sites. The normalized peak areas were corrected for the natural abundance using the R package Iso-CorrectoR (Heinrich et al, 2018).

For the analysis of anionic compounds by AEC-MS, samples were diluted with MilliQ water (1:2 vol/vol). Measurements were performed using combination of a Dionex ICS-6000 HPIC and a high field Thermo Fisher Scientific Q Exactive Plus quadrupole-Orbitrap mass spectrometer (both Thermo Fisher Scientific) as described earlier with minor modifications (Curien et al, 2021). 10 $\mu l$ of sample was injected via a Dionex AS-AP autosampler in push partial mode. Anion exchange chromatography was conducted on a Dionex IonPac AS11-HC column (2 mm × 250 mm, 4 $\mu m$ particle size, Thermo Fisher Scientific) equipped with a Dionex IonPac AG11-HC guard column (2 mm × 50 mm, 4 $\mu m$, Thermo Fisher Scientific) at 30°C. The mobile phase was established using an eluent generator with a potassium hydroxide cartridge to produce a potassium hydroxide gradient. The column flow rate was set to 380 $\mu l\ min^{-1}$ with a starting KO-H concentration of 5 mM for 1 min. The concentration was increased to 85 mM within 35 min and held for 5 min. The concentration was immediately reduced to 5 mM and the system equilibrated for 10 min. Spray stability was achieved with a makeup consisting of methanol with 10 mM acetic acid delivered with 150 $\mu l\ min^{-1}$ by an AXP Pump. The electro spray was achieved in the ESI source using the following parameters: sheath gas 30, auxiliary gas 15, sweep gas 0, spray voltage—2.8 kV, capillary temperature 300°C, S-Lens RF level 45, and auxiliary gas heater 380°C. For the untargeted approach, the mass spectrometer operated in a combination of full mass scan and a data-dependent Top5 MS2 (ddMS2) experiment. The full scan (60–800 m/z) was conducted with a resolution of 140,000 and an automatic gain control (AGC) target of $10^6$ ions with a maximum injection time of 500 ms. The Top5 ddMS2 experiment was carried out with a resolution of 17,500 and an AGC target of $10^5$ and a maximum IT of 50 ms. The stepped collision energy was used with the steps (15, 25, 35) to create an average of NCE 25. Data analysis was conducted using Compound Discoverer (version 3.1; Thermo Fisher Scientific) using the "untargeted Metabolomics workflow" for steady-state analysis. Compound identification was achieved on the level of mass accuracy (MS1 level), fragment mass spectra matching (MS2 level) and retention time comparison with authentic standards. For the enrichment analysis with stable heavy isotopes, the standard workflow for "stable isotope labelling" was chosen with the default settings 5 ppm mass tolerance, 30% intensity tolerance and 0.1% intensity threshold for isotope pattern matching and a maximum exchange rate was of 95%.

For free fatty acid analysis via GC-MS, lipids were hydrolysed and free fatty acids were methylated to fatty acid methyl esters (FAMEs). To do so, the organic phase was transferred into a glass vial and dried under a stream of nitrogen gas. The dried sample was resuspended in 1 ml of methanolic hydrochloride (MeOH/3 N HCl) and incubated at 90°C for 1 h. One mL of hexane and 1 ml of NaCl solution (1%) were added before centrifugation at 2,000$g$ for 5 min. The FAME-containing organic phase (top layer) was collected in a clean glass vial and stored at −20°C until measurement as described recently (Vasilopoulos et al, 2024).

## Quantification of mitochondrial morphology, membrane potential ($\Delta\Psi_m$), and cellular lipid droplets

HepG2 cells (0.25 × $10^6$ cells) were seeded onto 35 mm Poly-D-Lysine-coated (50 $\mu g/ml$) live-imaging dishes (P35G-1.5-14-C; MATTEK) and incubated for 24 h at 37°C, 5% $CO_2$ in the corresponding normoglycemic or hyperglycemic media. The assessment of mitochondrial morphology, $\Delta\Psi_m$ and cellular lipid droplets was performed by addition of MitoTracker Green (200 nM; Invitrogen), TMRM (50 nM; Invitrogen), BODIPY 493/503 (10 $\mu M$; Cayman Chemicals), respectively, for 30 min at 37°C, followed by washing thrice. Live-cell microscopy was performed using a spinning disc confocal microscope (PerkinElmer) equipped with a 60x oil-immersion objective (N.A = 1.49) and a Hamamatsu C9100 camera (1,000 × 1,000 pixel). The cells were maintained at 37°C in DMEM supplemented with 10 mM Hepes for the imaging duration. MitoTracker Green and BODIPY 493/503 were excited with a 488 nm laser, whereas TMRM was excited with a 561 nm laser. The images were obtained at emission wavelength of 527 nm (W55) and 615 nm (W70) for 488 nm and 561 nm excitation, respectively. The cell population was classified into tubular, intermediate, and fragmented mitochondrial morphology based on most of the mitochondria belonging to the respective class. Cells classified as tubular and fragmented contained mostly long tubular and short fragments, respectively, whereas cells classified as intermediate had a mixture of mostly short pieces, few long tubes, and fragmented mitochondria. Volocity image analysis software was used for the quantification regarding $\Delta\Psi_m$ and lipid droplets. The total fluorescence intensities of TMRM and BODIPY were obtained per cell after respective background subtraction. Each cell was manually demarcated by drawing a ROI. Lipid droplet number within a ROI was obtained automatically using find spots by setting threshold of brightest spot within a radius of 0.5 $\mu m$ and compartmentalization to ROI.

## Glucose uptake assay

3 × $10^4$ HepG2 cells were seeded in triplicates onto a dark 96-well plate overnight and in parallel onto a clear 96-well plate for cell normalization. Cellular glucose uptake was measured using Glucose Uptake-Glo Assay kit (Promega), according to the manufacturer's protocol. Luminescence was measured by microplate reader (CLARIOstar Plus, BMG LABTECH) with 1 s integration after 1 h of incubation. Normalization was performed using Hoechst staining and mean of signal intensity was used for normalizing luminescence intensities. Luciferase signals were normalized to WT-N measurement.

## Mitochondrial respirometry

A variety of respirometry experiments were performed using Seahorse XFe96 Analyzer (Agilent). HepG2 cells were seeded onto Poly-D-Lysine-coated (50 μg/ml) Seahorse XF96 cell culture plate (Agilent) at a density of $3.0 \times 10^4$ cells per well. For mitochondrial stress test, mitochondrial fuel flexibility test and glycolysis stress test, cells were incubated overnight in standard growth media. For fatty acid oxidation (FAO) test, standard growth medium was replaced by serum-deprived growth medium (DMEM without glucose, pyruvate and glutamine), containing 1% FBS, 0.5 mM glucose, 0.5 mM L-Carnitine (Sigma-Aldrich) and 1.0 mM glutamine 10 h after cell seeding and incubated overnight.

Before performing the assay, old medium was removed and cells were washed twice after which cells were supplemented with the corresponding assay media followed by 45 min $CO_2$-free incubation. Mitochondrial stress test was performed using Seahorse assay media (Agilent) supplemented with 10 mM glucose, 2 mM stable glutamine, and 1 mM sodium pyruvate. Mitochondrial oxygen consumption was measured after sequential addition of oligomycin (1 μM), FCCP (0.25 μM), and rotenone/antimycin (0.5 μM) according to the manufacturer's protocol. Mitochondrial fuel flexibility test was performed using Seahorse assay media containing 10 mM glucose, 2 mM stable glutamine, and 1 mM sodium pyruvate. After initial acquisition of basal respiration, glucose, glutamine, and FAO dependency and capacity was assessed according to manufacturer's protocol by sequential incubation with UK5099 (2 μM) and Etomoxir (4 μM)/BPTES (3 μM), BPTES (3 μM), and Etomoxir (4 μM)/UK5099 (2 μM) or Etomoxir (4 μM) and UK5099 (2 μM)/BPTES (3 μM), respectively. Glycolysis stress test was performed in Seahorse assay media supplemented with 2 mM glutamine. After 15 min of basal ECAR determination, glycolysis was induced by addition of glucose (10 mM), followed by oligomycin (1 μM) and lastly 2-DG (50 mM). For assessment of FAO, cells were pretreated with Seahorse assay media containing BSA (200 μM; Biomol) or Palmitate (200 μM; Biomol). FAO was measured by sequential addition of etomoxir (4 μM; Sigma-Aldrich) or media and mitochondrial stress test kit chemicals oligomycin (1.5 μM), FCCP (1 μM), rotenone/antimycin A (0.5 μM). Cell numbers were normalized using Hoechst (10 μg/ml) staining intensity assessed by microplate reader (M200 pro; Tecan). Data were analyzed using wave software (Agilent) and Microsoft Excel.

## Electron microscopy

$4 \times 10^6$ HepG2 cells were grown overnight in 10 cm petri dishes at 37°C with 5% $CO_2$ in the corresponding treatment media. Cells were fixed using 3% glutaraldehyde, 0.1 M sodium cacodylate buffer at pH 7.2 and subsequently pelleted. Cell pellets were washed in fresh 0.1 M sodium cacodylate buffer at pH 7.2 and embedded in 3% low melting agarose. Cells were stained using 1% osmium tetroxide for 50 min, washed twice with 0.1 M sodium cacodylate buffer and once using 70% ethanol for 10 min each. Thereafter, cells were stained using 1% uranyl acetate/1% phosphotungstic acid in 70% ethanol for 1 h. Stained samples were embedded in spur epoxy resin for polymerization at 70°C for 24 h. Ultrathin sections were prepared using a microtome and imaged on a transmission electron microscope (H600; Hitachi) at 75 V equipped with Bioscan 792 camera (Gatan). Image analysis was performed using ImageJ software.

## Sulforhodamine B (SRB) assay

Cell viability was assessed by SRB colorimetry assay. $2.5 \times 10^4$ HepG2 cells were seeded in 24 well plates and incubated for 24, 48 or 72 h. Subsequently, cells were washed with PBS and fixed with 10% (wt/vol) cold trichloroacetic acid solution (500 μl/well) for 1 h at 4°C. After washing five times with MilliQ water, cells were dried at RT overnight. Fixed cells were stained with SRB solution (0.4% [wt/vol] in 1% acetic acid, 300 μl/well) for 15 min at RT, washed five times with 1% acetic acid and dried at RT for 1 h. SRB extraction was performed by addition of 400 μl Tris-Base (10 mmol/liter) per well. The absorbance was measured, after 5 min of shaking, at 492 nm and 620 nm using a microplate reader (M200 pro; Tecan). Total intensity was calculated from signal intensity at 492 nm after background subtraction of 620 nm intensity. Proliferation was normalized to WT-N.

## Statistics and data representation

Data are represented as mean ± SEM. Statistical significance was determined by one-way ANOVA followed by Šídák's test for multiple comparisons of selected pairs with *$P$-value ≤ 0.05, **$P$-value ≤ 0.01, ***$P$-value ≤ 0.001, ****$P$-value ≤ 0.0001. Data analysis was performed using Microsoft Excel (Tables S1, S2, S3, S4, and S5). Data representation and statistical analysis was performed using GraphPad Prism.

# Data Availability

The mass spectrometry proteomics data from this publication have been deposited to the PRIDE partner repository and assigned the identifier PXD047246. The transcriptomics data from this publication have been deposited to the NCBI's Gene Expression Omnibus (GEO) database and assigned the GEO series accession number GSE248848.

# Supplementary Information

# Acknowledgements

AK Kondadi received Deutsche Forschungsgemeinschaft (DFG, German Research Foundation) Grant KO 6519/1-1. AS Reichert received grants from DFG Graduiertenkolleg VIVID RTG 2576 and DFG SFB 1208, project B12 (ID 267205415). R Anand received DFG Grant AN 1440/3-1. We thank Gisela Pansegrau and Andrea Borchardt for technical assistance in Western blot and electron microscopy experiments, respectively. We are thankful for excellent technical support by Maria Graf, Elisabeth Klemp, and Katrin Weber (CMML). RNA-sequencing was performed at the Genomics and

Transcriptomics facility, Biological and Medical Research Center (BMFZ), Medical Faculty, Heinrich-Heine-University Düsseldorf. Computational infrastructure and support were provided by the Centre for Information and Media Technology at Heinrich Heine University Düsseldorf. Proteome experiments were performed at the proteomics facility, BMFZ, HHU, Düsseldorf. Metabolite analyses were supported by the CEPLAS Plant Metabolism and Metabolomics Laboratory, which is funded by the DFG under Germany's Excellence Strategy—EXC-2048/1—project ID 390686111.

## Author Contributions

M Damiecki: data curation, formal analysis, validation, investigation, visualization, methodology, and writing—original draft, review, and editing.

R Naha: formal analysis and investigation.

Y Schaumkessel: formal analysis and investigation.

P Westhoff: formal analysis, investigation, and methodology.

N Atanelov: formal analysis and investigation.

A Stefanski: formal analysis, investigation, and methodology.

P Petzsch: formal analysis, investigation, and methodology.

K Stühler: resources.

K Köhrer: resources.

APM Weber: resources and funding acquisition.

R Anand: resources, funding acquisition, and writing—review and editing.

AS Reichert: conceptualization, resources, funding acquisition, project administration, and writing—review and editing.

AK Kondadi: conceptualization, supervision, funding acquisition, visualization, project administration, and writing—original draft, review, and editing.

## Conflict of Interest Statement

The authors declare that they have no conflict of interest.

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
