## [Reviewer comments · Life Science Alliance]

Mitochondrial Apolipoprotein MIC26 is a metabolic rheostat regulating central cellular fuel pathways

Melissa Damiecki, Ritam Naha, Yulia Schaumkessel, Philipp Westhoff, Nika Atanelov, Anja Stefanski, Patrick Petzsch, Kai Stühler, Karl Köhrer, Andreas P M Weber, Ruchika Anand, Andreas S Reichert and Arun Kumar Kondadi
DOI: 10.26508/lsa.202403038

Corresponding author(s): Dr. Arun Kumar Kondadi (Heinrich Heine University Düsseldorf)

Review timeline:

Submission Date:	2024-09-10
Editorial Decision:	2024-09-12
Revision Received:	2024-09-22
Accepted:	2024-09-23

Scientific Editor: Eric Sawey

Transaction Report:

Please note that the manuscript was reviewed at Review Commons and these reports were taken into account in the decision-making process at Life Science Alliance.

Review
COMMONS

Manuscript number: RC-2024-02345

Corresponding author(s): Arun Kumar, Kondadi

1. General Statements [optional]

This section is optional. Insert here any general statements you wish to make about the goal of the study or about the reviews.

This section is mandatory. Please insert a point-by-point reply describing the revisions that were already carried out and included in the transferred manuscript.

Reviewer #1 (Evidence, reproducibility and clarity (Required)):

Summary:

MIC26 is a subunit of the 'mitochondrial contact site and cristae organizing system' (MICOS) complex required for crista junction (CJ) formation and was functionally linked to diabetes and modulation of lipid metabolism. In order to understand the role of MIC26 in metabolism, the authors generated MIC26-KO HepG2 cells and investigated the pathways regulated by MIC26 under normo- and hyper-glycemic culture conditions. They employed a multi-omics approach that include transcriptomics, proteomics, targeted metabolomics, and functional assays to document the changes in mRNAs, proteins, and metabolites as a result of MIC26 deletion. Through bioinformatic analyses, they showed that the function of MIC26 is critical in various pathways regulating fatty acid synthesis, oxidation, cholesterol metabolism, and glycolysis. Interestingly, they found an entirely antagonistic effect of lipogenesis in MIC26-KO cells compared to WT cells depending on the glucose concentration of the culture media. In addition, they showed that MIC26 deletion led to a major metabolic rewiring of glutamine utilization as well as oxidative phosphorylation.

Major comments:

1) This is basically a descriptive study that document the transcriptomic, proteomic, and metabolic consequences of lacking MIC26 in normal or high glucose environment. It is data rich but insight poor. The connections between MIC26 as a subunit of MICOS complex and all those metabolic pathways are so tenuous that it is hard to see what to follow up after.

Response: We respectfully differ from the reviewer's opinion that the manuscript is data rich and insight poor. Our study provides significant insights by demonstrating how MIC26, strategically residing in the mitochondrial inner membrane (IM), regulates major cellular pathways.

MIC26 operates in a dual manner:

- A) Depending on the nutritional status, normoglycemia or hyperglycemia, MIC26 regulates the glycolysis, lipid, cholesterol metabolism and TCA cycle intermediates in an antagonistic manner.
- B) Independent of the nutritional status, it regulates glutamine and OXPHOS metabolism.

Full Revision

In addition, based on the suggestions by Reviewer #2, we have tested whether other proteins of MICOS (MIC27 and MIC19) present in two different sub-complexes regulate important metabolic pathways. Using the experimental results achieved (See reply to comments from Reviewer #2), we conclude that MIC26 plays a unique role as metabolic regulator in the IM and this study is therefore important to the general field of metabolism.

2) In the Results section (page 5, line 114-123), the description of the Western blot (WB) analysis appears inconsistent with several blot images of Fig. 1A, which makes the result unconvincing. The authors should select appropriate representative WB images, assuming they have them, to support their claim.

Response: Thank you for the suggestion. Firstly, we have performed additional experiments in this regard, and included the relevant quantification. Secondly, we have replaced some WBs which depict the appropriate quantification.

Further, we also modified the relevant lines in the manuscript stating that 'Mitochondrial apolipoproteins, MIC26, MIC27, and MIC25 are increased in cells exposed to hyperglycemia' and not only MIC26 as stated before.

Minor comments:

3) As the functional role of MIC26 in metabolism is the primary focus, the authors should present the results in the figures in the order of WT-N, MIC26KO-N, WT-H, MIC26KO-H for easier comparison.

Response: We understand the reasoning to interchange the two conditions. However, such an endeavour will involve cropping images of WBs, BN-PAGE and Clear native PAGE to better represent the corresponding quantification. This will also involve modifying all figures (data-sets and functional assays) in the whole manuscript. Overall, considering that the benefits of interchanging the order, of the WT-Hyperglycemia and MIC26 KO-Normoglycemia, are relatively minor, we have decided to stick to the original representation of the figures.

Reviewer #1 (Significance (Required)):

Mitochondria play important roles in metabolism and metabolic disorders. This study generated large amount of data relating to the role of a mitochondrial protein MIC26 in metabolism. Mutations in MIC26 have been associated with mitochondrial myopathy, lactic acidosis, and cognition defects (Beninca et al. 2021) and a lethal progeria-like condition (Peifer-Weib et al. 2023). There is also a connection between MIC26 and metabolic disorders. The results of this study will be of interest to researchers in the fields of mitochondrial diseases and metabolic disorders.

My field of expertise is mitochondrial disease, proteomics, lipidomics, phospholipid biochemistry.

Reviewer #2 (Evidence, reproducibility and clarity (Required)):

Summary

This study determines the role of MIC26, a mitochondrial component of the MICOS complex, in influencing cellular metabolic status. Using a variety of multiomic profiling techniques, as well as functional assays such as Seahorse-based respirometry, the authors propose that MIC26 regulates a variety of metabolic processes, including lipid and cholesterol homeostasis, glycolysis, fatty acid oxidation, fatty acid synthesis, TCA cycle homeostasis, glutamine metabolism, and general mitochondrial bioenergetics via OxPhos activity and supercomplex formation. The data were generated in MIC26 knockout HepG2 hepatocellular carcinoma cells grown in two nutrient conditions: high glucose (which they term "hyperglycemia") and low glucose (which they term "normoglycemia") DMEM. While the data support the authors' conclusion that loss of MIC26 causes broad metabolic changes across these conditions, the authors do not distinguish whether MIC26 knockout affects metabolism due to its canonical role in MICOS, or whether it acts as "a metabolic rheostat" to directly regulate central cellular fuel pathways, as is claimed in the title of this manuscript. Given the breadth of metabolic alterations seen in the MIC26 KO cells, it seems likely that at least a subset of these changes are indirect, rather than that MIC26 plays a direct regulatory role in the eight distinct metabolic pathways outlined above. Thus, while the data generally support the conclusion that expression of MIC26 is important for metabolic homeostasis, the mechanism(s) by which MIC26 influences cellular metabolism remains unclear and should be further addressed.

Major Comments

1) The authors infer that MIC26 influences cellular metabolism by referencing a handful of papers on MIC26 transcript differences in select metabolic models, metabolic alterations seen in a MIC26 transgenic/overexpression mouse model, or metabolic effects seen due to MIC26 tissue specific KO models. They thus hypothesize that "MIC26 has an unidentified regulatory role under nutrient-enriched conditions" (line 88). They support this observation by showing that MIC26 is increased in high glucose DMEM relative to low glucose DMEM in Figure 1. However, the authors claim, in both the figure legend title as well as the results section header, that "MIC26 is selectively increased in cells exposed to hyperglycemia" (lines 103-4), when their data demonstrate that this is not true. Figure 1B shows that MIC27, MIC26, and MIC25 increase in high glucose relative to low glucose conditions, indicating that MIC26 is not selectively increased, but rather that multiple subunits of MICOS are increased under nutrient-enriched conditions.

Response: We agree to the reviewer's comment and have now replaced the title in the results section in the manuscript accordingly - 'Mitochondrial apolipoproteins, MIC26, MIC27, and MIC25 are increased in cells exposed to hyperglycemia' and not only MIC26 as stated before. Further, in order to strengthen the WB data from Fig 1A & B, we also increased the number of experiments and updated the Fig 1B and also some WBs in Fig 1A to better represent the quantification.

2) This does not suggest that MIC26 does not have an important role in maintaining metabolism under such nutrient conditions, but rather supports a model in which MICOS itself dynamically responds to altered nutrient conditions in cell culture. Interestingly, other MICOS subunits do not change in abundance (e.g., MIC19 and MIC60), which have previously been associated with a separate MICOS subcomplex than MIC26 in yeast (PMID: 33053165). These data may suggest that the nutrient

responsive behavior of MIC26 may be due to its assembly within this specific MICOS subcomplex, rather than an independent "unidentified regulatory role under nutrient-enriched conditions".

To test this, the authors should repeat a subset of functional assays (perhaps the Seahorse metabolic assays) in other MICOS deletion cell lines, including one that dynamically changes in expression in a similar manner to MIC26 (e.g., MIC27 KO or MIC25 KO) and one that does not dynamically change in expression in high glucose conditions (e.g., MIC60 or MIC13). In these sets of experiments the authors will be able to distinguish three possibilities:

Model 1: if MICOS in general affects metabolic pathways, similar results will be seen for all MICOS subunit KOS.

Model 2: if the MIC26-specific subcomplex is dynamically regulated to influence cellular metabolism, KO of MIC26 and other subunits of this subcomplex should show similar results, but KO of subunits of the non-dynamic subcomplex (e.g., MIC60) would not show similar phenotypes.

Model 3: If only MIC26 KO, but not other MICOS subunits, show metabolic phenotypes, this would support a MICOS-independent role for MIC26 in influencing cellular metabolism.

Importantly, any of these models are interesting, and testing these models does not invalidate any of the phenotypes presented in the manuscript. Rather, these experiments would assist the reader in understanding the underlying mechanism by which MIC26 loss causes cellular metabolic defects. Furthermore, it is worth stating that performing all experiments with multiple other MICOS cell lines is beyond the scope of the manuscript, but testing effects in select (preferably functional) assays, such as the glycolysis stress test (Fig 3E), FAO Seahorse (Fig 3H-J) glutamine oxidation (Fig 6B), and general stress test (Fig 7E) would be appropriate. Other more defined and easily achievable experiments could also be used to support these claims (e.g., western blots probing for levels of key metabolic regulators).

Response: We appreciate the balanced comments of the reviewer who has carefully read and appreciated our manuscript. We also appreciate the constructive criticism of the reviewer who suggested various possible models considering the MIC26 role.

In this endeavour, we have performed the following extensive experiments:

- 1) Generated *MIC19* KOs in HepG2 cells. We had already generated *MIC27* KOs during the course of a previous publication (Lubeck et al, 2023) and used them for this publication (**Fig S4**). WB analyses was performed using the *MIC27* KO and *MIC19* KO cells.
- 2) Measured glycolysis function using the glycolysis stress test in *MIC27* and *MIC19* KOs (**Fig S5**).
- 3) Analysed lipid metabolism by imaging and extensively quantifying LD number and BODIPY fluorescence intensities in *MIC27* and *MIC19* KO cells (**Fig S6**).
- 4) Measured glutamine oxidation using Mito Flex fuel tests in *MIC27* and *MIC19* KOs (**Fig S10F**).
- 5) Analysed general mitochondrial respiration by using mito stress kit in *MIC27* and *MIC19* KOs (**Fig S11-E-H**).

1) Characterisation of *MIC27* and *MIC19* KO cells, using WB analyses, cultured in normoglycemia and hyperglycemia

New Figure S4

(A) Representative WBs of all MICOS subunits from HepG2 WT, *MIC27* KO and *MIC19* KO cells cultured in normo- and hyperglycemia. (B) Quantification of all MICOS subunits from respective cells (N = 3-5). Data are represented as mean ± SEM (B). Statistical analysis was performed using one-way ANOVA with **P* < 0.05, ***P* < 0.01, ****P* < 0.001, *****P* < 0.0001 for all meaningful combinations (normoglycemia to hyperglycemia for each genotype and KO to WT for each treatment condition). Non-significant *P* values are not shown. N represents the number of biological replicates.

2) Glycolysis stress kit

New Figure S5

(A) Representative glycolysis stress test seahorse assay analysis, with sequential injection of glucose, oligomycin and 2-deoxyglucose, reveals a tendency towards increased glycolysis upon *MIC27* and *MIC19* deletion (n = 15).

(B – D) Respective quantification of glycolytic reserve (B), glycolytic function (C) and glycolytic capacity (D), from various biological replicates (N = 3).

Data are represented as mean ± SEM (A-M). Statistical analysis was performed using one-way ANOVA with **P* < 0.05, ***P* < 0.01, ****P* < 0.001, *****P* < 0.0001 for all meaningful combinations (normoglycemia to hyperglycemia for each genotype and KO to WT for each treatment condition). Non-significant *P* values are not shown. N represents the number of biological replicates and n the number of technical replicates.

3) Analysis of lipid droplet parameters

New Figure S6

(A – D) Analysis of lipid droplet formation in WT, *MIC27* KO and *MIC19* KO cells cultured in normo- and hyperglycemia either in standard growth condition (CTRL) or upon palmitate stimulation (100 µM, 24 h). Representative confocal images of lipid droplets stained using BODIPY 493/503 are shown (A).

Quantification shows number of lipid droplets normalized to the total cell area [μm^2] (B) and mean fluorescence intensity per cell normalized to mean intensity of WT-N in all biological replicates (C) ($N = 3$). Scale bar represents $5 \mu\text{m}$.

Data are represented as mean \pm SEM (A-M). Statistical analysis was performed using one-way ANOVA with $*P < 0.05$, $**P < 0.01$, $***P < 0.001$, $****P < 0.0001$ for all meaningful combinations (normoglycemia to hyperglycemia for each genotype and KO to WT for each treatment condition). Non-significant P values are not shown. N represents the number of biological replicates and n the number of technical replicates.

4) Glutamine Metabolism

F

New Figure S10F

(F) Quantification of mitochondrial glutamine dependency and capacity analysis, using Seahorse XF analyzer mito fuel flex test assay, show a dependency of *MIC27* KO and *MIC19* KO cells on glutamine fueled respiration ($N = 3$).

Data are represented as mean \pm SEM (A-D). Statistical analysis was performed using one-way ANOVA with $*P < 0.05$, $**P < 0.01$, $***P < 0.001$, $****P < 0.0001$ for all meaningful combinations (normoglycemia to hyperglycemia for each genotype and KO to WT for each treatment condition). Non-significant P values are not shown. N represents the number of biological replicates.

5) Steady-state respirometry using mito stress kit

New Figure S11 E-H

(E) Representative mitochondrial stress test with Seahorse XF analyzer, with sequential injection of oligomycin, FCCP and rotenone/antimycin (E) ($n = 15$). (F-H) Quantifications from various biological replicates shows an unaltered basal respiration (F), ATP production (G) and spare respiratory capacity (H) in *MIC27* KOs and *MIC19* KOs cultured in both normo- and hyperglycemia ($N = 3$).

Data are represented as mean \pm SEM (A-D). Statistical analysis was performed using one-way ANOVA with $*P < 0.05$, $**P < 0.01$, $***P < 0.001$, $****P < 0.0001$ for all meaningful combinations (normoglycemia to hyperglycemia for each genotype and KO to WT for each treatment condition). Non-significant P values are not shown. N represents the number of biological replicates.

We provide a summary of the above results:

#	Metabolic Pathway	Experiment	MIC26 KO	MIC27 KO	MIC19 KO
1	Glycolysis	Glycolysis stress kit	Glycolytic reserve increased	Glycolytic reserve unchanged	Glycolytic reserve unchanged
2A	Lipid Metabolism	LD number	General increase	No consistent increase in treatment with and without palmitate in normoglycemia and hyperglycemia	No consistent increase in treatment with and without palmitate in normoglycemia and hyperglycemia
2B	Lipid Metabolism	LD (BODIPY) intensities	Presence of antagonistic regulation of LD content	Presence of antagonistic regulation of LD content	Absence of antagonistic regulation of LD content
3	Glutamine oxidation	Mito flex fuel test	No dependency	Dependent	Dependent
4	Steady-state respiration	Mito stress test	Basal respiration increased	Basal respiration unchanged	Basal respiration unchanged
5	Steady-state respiration	Mito stress test	SRC decreased in normoglycemia	SRC unchanged	SRC unchanged

Taking the above summary, we investigated three possibilities considering the role of MIC26:

- 1) General role of MICOS – whether deletion of any MICOS protein leads to similar phenotype as *MIC26* deletion.
- 2) Specific role of MIC26/27/10 subcomplex – whether deletion of any other protein in the MIC26-subcomplex like MIC27 leads to similar results, accompanied by dissimilar results in KOs of any protein belonging to the other MICOS subcomplex (MI19/MIC25/MIC60) and whose protein levels were not changed upon hyperglycemia treatment (*MIC19* or *MIC60* KOs).
- 3) MICOS-independent role of MIC26.

Considering the various metabolic pathways analysed, we conclude that MIC26 has a MICOS-independent role in regulating major cellular pathways.

Minor Comments

1. The multiomics data as presented in Figure 2 is difficult to interpret. This is mainly driven by the fact that there are multiple comparisons that should be communicated (KO v WT, normoglycemia v hyperglycemia, upregulated v. downregulated), but only select enrichment values are shown (e.g., normoglycemia upregulated in 2C and hyperglycemia downregulated in 2D). It took me a long time as a reader to understand what I was looking at because only select analyses are presented. What pathways are upregulated in hyperglycemia in *MIC26* KO v. WT?

Response: Thank you for pointing this out. We have now represented all the four conditions regarding enrichment analysis as suggested. The antagonistic metabolic regulation is only observed in *MIC26* KO cells cultured in normoglycemia (upregulated) when compared to *MIC26* KOs cells cultured in hyperglycemia (downregulated). When *MIC26* KOs cultured in hyperglycemia (upregulated) were compared with *MIC26* KOs cultured in normoglycemia (downregulated), there were also few pathways which showed an antagonistic regulation, but not directly involved with metabolism, relating to apoptosis etc. Hence, we did not focus on these pathways in the current manuscript.

The Treemaps have been shifted to Fig S1. All four Treemaps have been represented instead of the two shown before. In the process, the previous proteomics Fig S1B and S1C depicting antagonism in different pathways have been excluded.

2. Figure 3 would be stronger if expression from all glycolytic proteins were shown instead of only a subset. If the authors are making the claim that MIC26 KO increases glycolytic flux via protein-level upregulation of glycolysis, this could be substantiated at a pathway level. These data could be included in supplemental data if they are difficult to fit into the figure.

Response: Thank you. We have now included the data of proteins regulating glycolysis as a new figure (Fig S3C). *MIC26* KO cells cultured in normoglycemia had increased levels of aldolase (ALDOA & ALDOC), phosphoglycerate kinase (PGK1) and pyruvate kinase (PKM & PKLR) when compared to control cells. We have included this data in the manuscript.

3. In Figure 4H-J - the raw data for the Seahorse traces should be shown, and OCR should be reported in pmol/min rather than relative percentages so as to help the reader more critically evaluate the data.

Response: For the FAO assays, we treat the cells with palmitate or mock (BSA serving as control). The histograms (Fig 4H-J) are represented as such because we normalised the oxygen consumption after palmitate treatment with oxygen consumption of mock-treated cells. We understand the reviewer's concern and have now included the absolute values of oxygen consumption of FAO assay in an excel sheet (Supplementary Table S5). In addition, we have also included the absolute values for mito stress test and glycolysis assays where the oxygen consumption has been normalised to WT-normoglycemia condition (Supplementary Table S5). The original oxygen consumption curves for glycolysis stress kit (Fig 3E, S5A) and mito stress kits (Fig 7E, S11E) are shown as figures.

4. The majority of the plots are shown with 4 comparisons, but statistical comparisons are often only provided for a subset of comparisons. It is unclear whether statistics were compared across all comparisons and the non-annotated comparisons are not significant, or whether those calculations were not performed. Defining this in the figure, or, better, annotating all relevant comparisons on each graph with "ns" for not significant, would assist the reader with interpreting the data.

Response: Thanks for pointing this out. After comparing all meaningful conditions (except WT-N to *MIC26* KO-H and WT-H to *MIC26* KO-N), only those that were significant were represented using asterisks. We have now mentioned this information in the respective figure legends. We avoided using 'non-significant (ns)' in the figure as it would make some figures very crowded as seen from some of our trials.

Reviewer #2 (Significance (Required)):

This study broadly profiles the metabolic defects associated with loss of the MICOS subunit MIC26 in hepatocellular carcinoma cells in variable nutrient conditions (e.g., high glucose and low glucose). As a reviewer with expertise in multiomic profiling of metabolic models, I found the breadth of pathways studied in this manuscript to be impressive. Furthermore, the authors use a variety of techniques, including multiomic profiling, isotopic flux analysis, and functional Seahorse assays to support their conclusions. The study provides a comprehensive analysis of metabolic changes associated with MIC26, and is thus an important advance in profiling how loss of MIC26 (or MICOS in general; see below) affects cellular metabolism in the context of dynamic nutrient changes. However, the claim that "MIC26 is a metabolic

Full Revision

rheostat regulating central cellular fuel pathways", as is proposed in the title of the manuscript, is unsubstantiated, as the authors do not test whether loss of MIC26 specifically influences cellular metabolism independent of its role in MICOS. This paper would be significantly strengthened if a subset of functional assays across metabolic pathways were repeated with other MICOS KO cell lines to delineate whether these metabolic effects are direct or indirect.

RE: Life Science Alliance Manuscript #LSA-2024-03038

Dr. Arun Kumar Kondadi
Heinrich Heine University Düsseldorf
Universitätstrasse 1
Düsseldorf, North Rhine Westphalia 40225
Germany

Dear Dr. Kondadi,

Thank you for submitting your revised manuscript entitled "Mitochondrial Apolipoprotein MIC26 is a metabolic rheostat regulating central cellular fuel pathways". We would be happy to publish your paper in Life Science Alliance pending final revisions necessary to meet our formatting guidelines.

- please be sure that the authorship listing and order is correct
- please upload your figures - both your main and supplementary figures - as single files
- please add a running title, summary blurb, and a category for your manuscript to our system
- please add the Twitter handle of your host institute/organization as well as your own or/and one of the authors in our system
- please consult our manuscript preparation guidelines <https://www.life-science-alliance.org/manuscript-prep> and make sure your manuscript sections are in the correct order
- the information in the file labeled "Figure Source Data" should instead be presented in a Data Availability statement placed at the end of the Materials and Methods section. The dataset deposited to PRIDE should also be made publicly accessible, removing the need for Reviewer access details.

LSA now encourages authors to provide a 30-60 second video where the study is briefly explained. We will use these videos on social media to promote the published paper and the presenting author (for examples, see <https://docs.google.com/document/d/1-UWCfbE4pGcDdcgzcmiuJI2XMBJnxKYeqRvLLrLS08s/edit?usp=sharing>). Corresponding or first-authors are welcome to submit the video. Please submit only one video per manuscript. The video can be emailed to contact@life-science-alliance.org

To upload the final version of your manuscript, please log in to your account: <https://lsa.msubmit.net/cgi-bin/main.plex>

A. FINAL FILES:

B. MANUSCRIPT ORGANIZATION AND FORMATTING:

Sincerely,

Eric Sawey, PhD

Executive Editor

Life Science Alliance

<http://www.lsjournal.org>

RE: Life Science Alliance Manuscript #LSA-2024-03038R

Dr. Arun Kumar Kondadi
Heinrich Heine University Düsseldorf
Universitätstrasse 1
Düsseldorf, North Rhine Westphalia 40225
Germany

Dear Dr. Kondadi,

Thank you for submitting your Research Article entitled "Mitochondrial Apolipoprotein MIC26 is a metabolic rheostat regulating central cellular fuel pathways". It is a pleasure to let you know that your manuscript is now accepted for publication in Life Science Alliance. Congratulations on this interesting work.

DISTRIBUTION OF MATERIALS:

Again, congratulations on a very nice paper. I hope you found the review process to be constructive and are pleased with how the manuscript was handled editorially. We look forward to future exciting submissions from your lab.

Sincerely,
